# *bfc*, a novel *serpent* co-factor for the expression of *croquemort*, regulates efferocytosis in *Drosophila melanogaster*

**Qian Zheng**[ORCID][☸], **Ning Gao**[☸], **Qiling Sun**[☸], **Xiaowen Li, Yanzhe Wang, Hui Xiao**[ORCID]*

College of Life Sciences, Shaanxi Normal University, Xi'an, China

☸ These authors contributed equally to this work.
* huixiao@snnu.edu.cn

**Data Availability Statement:** All RNAseq data files are available from the NCBI SRA database (accession number PRJNA733098). The URLs is https://www.ncbi.nlm.nih.gov/Traces/study/?acc=

## Abstract

Efferocytosis is the process by which phagocytes recognize, engulf, and digest (or clear) apoptotic cells during development. Impaired efferocytosis is associated with developmental defects and autoimmune diseases. In *Drosophila melanogaster*, recognition of apoptotic cells requires phagocyte surface receptors, including the scavenger receptor CD36-related protein, Croquemort (Crq, encoded by *crq*). In fact, Crq expression is upregulated in the presence of apoptotic cells, as well as in response to excessive apoptosis. Here, we identified a novel gene *bfc* (*booster for croquemort*), which plays a role in efferocytosis, specifically the regulation of the *crq* expression. We found that Bfc protein interacts with the zinc finger domain of the GATA transcription factor Serpent (Srp), to enhance its direct binding to the *crq* promoter; thus, they function together in regulating *crq* expression and efferocytosis. Overall, we show that Bfc serves as a Srp co-factor to upregulate the transcription of the *crq* encoded receptor, and consequently boosts macrophage efferocytosis in response to excessive apoptosis. Therefore, this study clarifies how phagocytes integrate apoptotic cell signals to mediate efferocytosis.

## Author summary

The swift removal of apoptotic cells by specialized cells such as macrophages (a subtype of white blood cells), is a critical event in shaping tissues during the development of all multicellular organisms, from worms to humans. Defects in the removal of dying cells, a process known as the clearance of apoptotic cells (ACs), can contribute to the development of autoimmune disorders such as systemic Lupus erythematosus (SLE) and neurodegenerative diseases like Alzheimer's disease. Croquemort (Crq), a *Drosophila* CD36-related receptor, is required for the recognition and engulfment of ACs in macrophages. In this study, via transcriptomic analysis and RNAi screening, we discovered 12 genes required for the clearance of ACs in macrophage-like S2 cells in *Drosophila*. In particular, we identified a novel gene, *bfc* (*booster for croquemort*), involved in apoptotic cell clearance via the specific regulation of the *crq* transcriptional expression. We demonstrate that the GATA transcription factor Serpent (Srp) directly binds to the *crq* promoter, while Bfc

PRJNA733098 All other data is available within the paper and its Supporting Information files.

**Funding:** H.X. received grants 91954114 and 31671439 from the National Natural Science Foundation of China (http://www.nsfc.gov.cn).Q.Z. received grant 31801164 from the National Natural Science Foundation of China (http://www.nsfc.gov.cn). H.X. received grants GK202001004 and GK202007009 from Shaanxi Normal University (http://www.snnu.edu.cn/). The funders had no role in study design, data collection and analysis, decision to publish, or preparation of the manuscript.

**Competing interests:** The authors have declared that no competing interests exist.

strengthens this binding via its interaction with the Srp zinc finger domain. Therefore, we propose a model in which Bfc cooperates with Srp to enhance the expression of *crq* and subsequently induce apoptotic cell clearance in *Drosophila melanogaster*.

## Introduction

Apoptosis is a developmentally programmed cell death process in multicellular organisms essential for the removal of excessive or harmful cells [1]; whereby apoptotic cells (ACs) are swiftly removed by phagocytes to prevent the release of toxins and induction of inflammation [2], a process crucial for organ formation, tissue development, homeostasis, and normal immunoregulation. In fact, defects in AC clearance (efferocytosis) [3,4] can lead to the development of various inflammatory and autoimmune diseases [5]. During efferocytosis, the effective clearance of ACs is accomplished through the recognition and binding of engulfment receptors or bridging molecules on the surface of phagocytes to "eat me" signals exposed on the surface of ACs [6–10]. After receptor activation, downstream signals trigger actin cytoskeleton rearrangement and membrane extension around the ACs to form phagosomes. Finally, mature phagosomes fuse with lysosomes to form phagolysosomes, where the internalized ACs are ultimately digested and cleared [11,12].

Since efferocytosis is conserved throughout evolution, it has been studied not only in mammals but also in *Drosophila melanogaster*. Of note, in *D. melanogaster*, ACs are removed by non-professional phagocytes, such as epithelial cells [13] and professional phagocytes, such as macrophages and glial cells [4]. Importantly, *Drosophila* macrophages perform similar functions to those of mammalian macrophages; they participate in both the phagocytosis of ACs and pathogens [4,14]. Several engulfment receptors have been identified as key players in the recognition and removal of ACs in *Drosophila*. Franc and colleagues first characterized Croquemort (Crq), a *Drosophila* CD36-related receptor required by macrophages to engulf ACs [8,15]. Additionally, Draper (Drpr, a homolog of CED-1/MEGF10) also mediates AC clearance in both glia and macrophages; JNK signaling plays a role in priming macrophages to rapidly respond to injury or microbial infections [16,17]. Of note, Drpr and its adapter Dmel\Ced-6 (GULP homolog) also seemed important for axon pruning and the engulfment of degenerating neurons by glial cells [18,19]. The Src tyrosine kinase Src42A (Frk homolog) promotes Drpr phosphorylation and its association with another soluble tyrosine kinase, Shark (ZAP70 homolog), which in turn activates the Drpr pathway [20]. In addition to Drpr, Six-Microns-Under (SIMU) [10] and integrin αPS3 [21] contribute to efferocytosis. SIMU, a Nimrod family cell surface receptor [22], functions upstream of Drpr to mediate the recognition and clearance of ACs as well as of non-apoptotic cells at wound sites [10,23] through the recognition of phosphatidylserine (PS). Importantly, the transcriptional factor Serpent (Srp), a GATA factor homolog [24,25], was recently found to be required for the efficient phagocytosis of ACs in the context of *Drosophila* embryonic macrophages and acted via the regulation of SIMU [10], Drpr, and Crq [26].

Searching for other genes required for efferocytosis, here we performed transcriptomic analysis (RNA-seq) and RNAi screening, and discovered 12 genes required for AC clearance in *Drosophila* S2 cells. In particular, we identified a novel gene, *bfc* (*booster for croquemort*), involved in efferocytosis that encodes a specific regulatory factor for the *crq* transcriptional expression, both *in vitro* and *in vivo*. Importantly, we demonstrate that the GATA factor Srp directly binds to the *crq* promoter, while Bfc strengthens this binding by interacting with the

Srp zinc finger domain. Therefore, we propose a model in which Bfc cooperates with Srp to enhance *crq* expression and subsequently induce efferocytosis in *D. melanogaster*.

## Results

### RNA-seq discloses 48 genes with expression patterns comparable to that of *crq* in S2 cells co-incubated with ACs

The expression of *crq* increases when apoptosis begins, whereas the absence of ACs results in decreased Crq levels [8]. To confirm this, we used *Drosophila* S2 cells (phagocytic cells with macrophage-like lineage derived from late embryonic stages) [27] in the presence or absence of ACs as an *in vitro* model of efferocytosis and measured the relative expression of *crq*. The ACs were prepared as previously described [28]; we confirmed that ACs were successfully obtained via Annexin V/PI staining (**S1A and S1B Fig**) and the detection of cleaved Dcp1 (**S1C Fig**). Of note, we analyzed the *crq* mRNA levels in S2 cells incubated with ACs over a course of 24 h (2, 4, 6, 8, 10, 12, 18, and 24 h; *versus* 0 h S2 cells without ACs as the control), and found that the expression of *crq* increased significantly at 4 h, remained high after 8 h, dropped sharply at 10 h, and remained at relatively low levels for the remaining time (**Fig 1A**). Therefore, we chose three time points (0, 6, and 12 h) to detect the Crq protein levels. Western blotting showed that the trend in Crq protein accumulation matched that of the above-mentioned transcriptional patterns in S2 cells co-incubated with ACs (**Fig 1B and 1C**). Importantly, the decreased Crq protein expression after 12 h incubation with ACs did not significantly affect the phagocytic activity of S2 cells; in fact, efferocytosis at 12 h was higher than that in S2 cells incubated with ACs for only 6 h (**Fig 1D and 1E**). Overall, our results show that the expression of Crq is induced by ACs at both the transcriptional and translational levels.

Since Crq is a well-known engulfment receptor, it is reasonable to consider that genes with expression patterns comparable to that of *crq* expression in S2 cells-ACs co-cultures may also be associated with efferocytosis. Therefore, next, we performed a transcriptomic analysis of S2 cells incubated with ACs for 6 h and 12 h and compared the results with those obtained using S2 cells not exposed to ACs using the Illumina HiSeq 2500 [29] platform. After filtering the low-quality reads and removing the rDNA or tRNA reads, we considered the genes with q < 0.05, and |log FC| ≥0.4 (FC = fold change) as differentially expressed. We obtained 1381, 1212, and 3620 differentially expressed genes from the comparisons performed at 6 h *versus* 0 h, 12 h *versus* 0 h, and 6 h *versus* 12 h, respectively (**S1 Data**). Interestingly, after setting the significance cut-off value to p< 0.05, and the threshold *crq* expression levels to > 1.8 or > 1.4 (6 h or 12 h, respectively *versus* 0 h), 50 genes were found to have a similar expression trend as of *crq* (**Fig 1F and 1G**), including pseudogene *CR43687*. Of note, S2 cells displayed changes in multiple transcriptional programs (**Fig 1F**), including the increased expression of innate immunity genes and genes involved in efferocytosis, in line with previously reported data [30].

### *CG9129* is involved in efferocytosis and regulates *crq* expression in S2 cells

To validate the RNA-seq data, we performed quantitative real time RT-PCR (qPCR) to test the status of representative genes selected across a range of expression levels (**Fig 1** and **S1 Table**). After qPCR validation in three independent experiments, we identified 24 genes whose expression patterns were highly similar to that of *crq* (**Fig 2A and 2B**). As necrosis may occur during the late stage of apoptosis, these genes may be upregulated by ACs or necrotic cells. Therefore, we performed RNAi suppression by individually targeting each gene in S2 cells (**S1D Fig**), and then incubated the RNAi treated cell lines with ACs to measure their efferocytosis capacity.

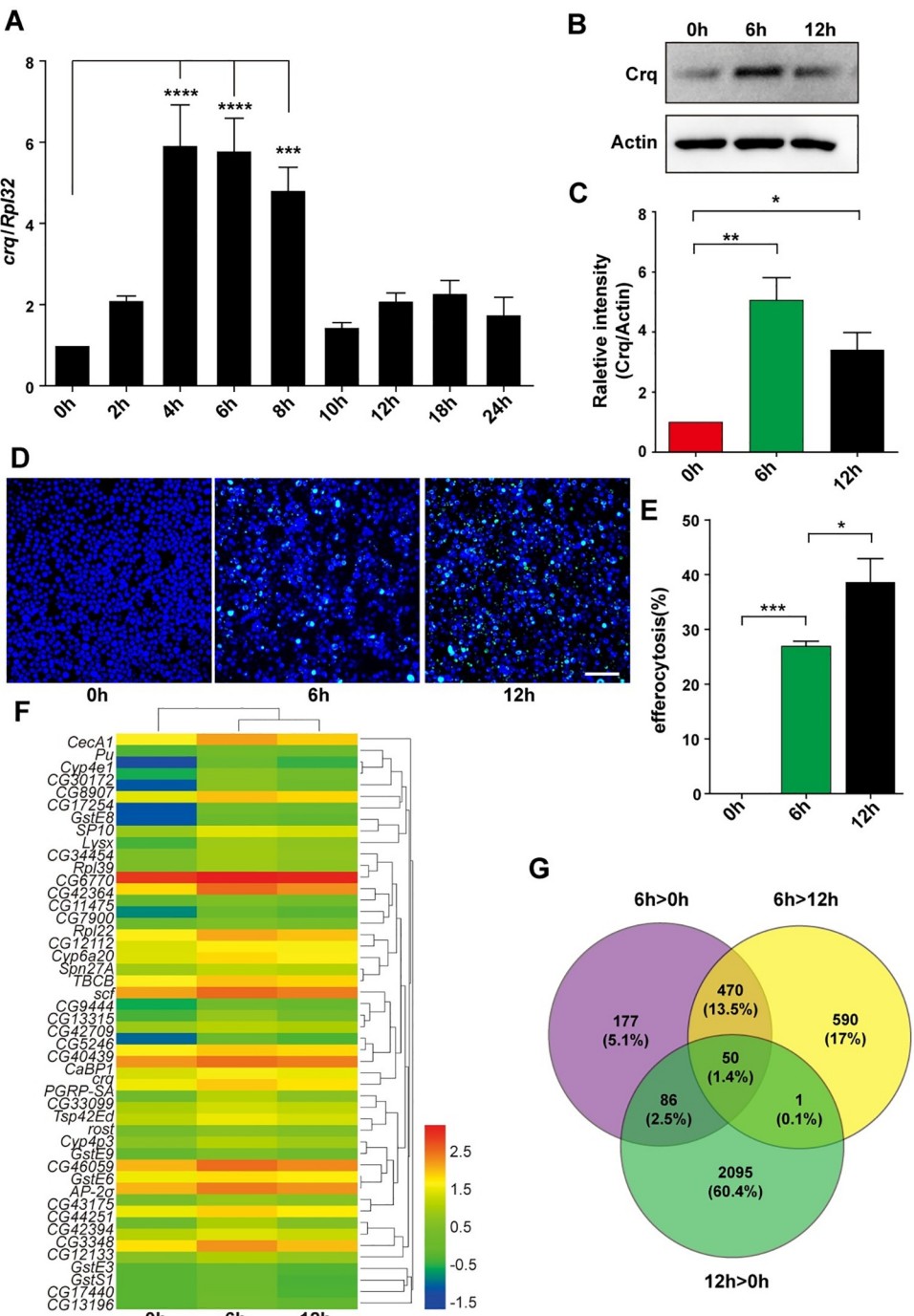

**Fig 1. The upregulated transcripts after the incubation of S2 cells with ACs. A**. The *crq* mRNA levels were quantified by quantitative RT-PCR, and normalized to those of *Rpl32* mRNA. **B.** Western blot analysis of Crq protein levels in S2 cells at 0, 6, and 12 h after the addition of ACs. Actin was used as the loading control. **C.** Quantification of the Crq protein levels shown in (**B**) after normalization to those of Actin, n = 3. **D.** AC efferocytosis by S2 cells at 0, 6, and 12 h. Live cells are shown in blue, and engulfed FITC-labeled ACs are shown in green. Scale Bars = 50 μm. **E.** Graph summarizing the quantification of the data in (**D**). The error bars represent the SEM. **F.** A heatmap of the 50 transcripts that showed similar expression patterns as of *crq*. Each column represents an independent sample. The color code and density represent the level of regulation: the red color defines genes with up-regulated expression, while the blue color defines genes with down-regulated expression. **G.** Venn diagram showing the differentially regulated transcripts after the addition of ACs: expression levels of 783, 1111 and 2232 transcripts were up-regulated in 6 h *versus* 0 h, 6 h *versus* 12 h, and 12 h *versus* 0 h comparisons, respectively. Statistical analysis was performed using the one-way ANOVA and the p values are denoted.

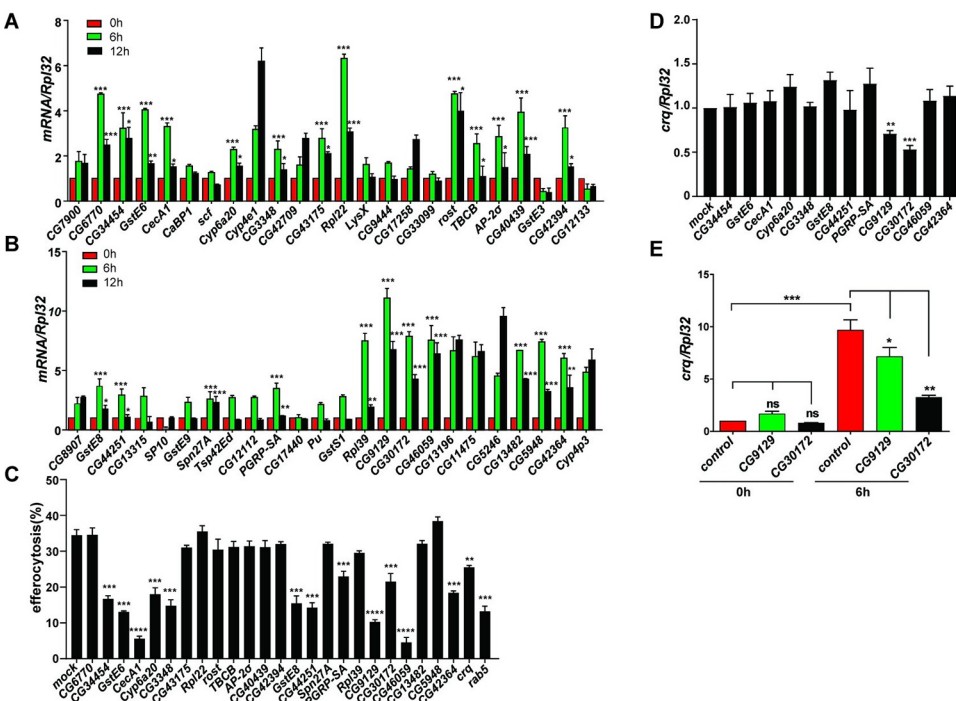

**Fig 2. CG9129 is involved in efferocytosis and regulates *crq* expression *in vitro*. A and B**. The transcription levels of differentially expressed genes determined by the transcriptomic analysis were validated by qPCR in S2 cells incubated with ACs for 0, 6, and 12 h. Twenty-four genes showed expression patterns similar to that of *crq*. The data were collected from three independent experiments; 0 h S2 cells were used as the control. The asterisks above the column of the 6 h time-point denote a significantly higher gene expression than that detected at 0 h. The asterisks above the column of the 12 h time-point denote a significantly lower gene expression than that detected at 6 h. **C**. Efferocytosis efficiency of RNAi-treated S2 cells after 6 h co-incubation with ACs. Each of the identified 24 genes was knocked down individually. The data are representative of three independent experiments. Significant differences were observed in comparison with mock RNAi-treated S2 cells (negative control), or *crq* and *Rab5* RNAi-treated S2 cells (positive controls). **D**. The *crq* mRNA levels were quantified by qPCR in RNAi-treated S2 cells showing decreased efferocytosis during 6 h incubation with ACs. The data are relative to the expression determined in mock RNAi-treated S2 cells. **E.** The relative *crq* mRNA levels were also quantified by qPCR in control, *CG9129* knockdown, and *CG30172* knockdown S2 cells in the absence (0 h) or presence of ACs (6 h incubation). Statistical analysis was performed using the two-way ANOVA (**A-B, E**) or one-way ANOVA (**C-D**).

Compared to mock RNAi treatments (34.4%) and the *Rab5* RNAi positive control (13.2%), 12 RNAi treatments led to reduced efferocytosis efficiency (**Fig 2C**).

We then assessed whether these 12 genes affected AC clearance via regulating the expression of *crq*. Interestingly, we found that RNAi treatments targeting *CG9129* and *CG30172* lead to decreased *crq* expression (by 30% and 47%, respectively; **Fig 2D**). Importantly, *CG9129*- and *CG30172*-targeted RNAi suppression did not affect the *crq* mRNA levels in S2 cells in the absence of ACs, indicating that *CG9129* and *CG30172* specifically regulate the expression of *crq* in response to stimulation by ACs (**Fig 2E**). Furthermore, the overexpression of CG9129 was sufficient to upregulate the *crq* mRNA expression in the presence of ACs (**S2A Fig**). Interestingly, Crq was also shown to regulate the transcription of *CG9129* (**S2 Fig**), suggesting a positive feedback mechanism between the transcription of *CG9129* and *crq* in efferocytosis. Of note, *CG9129* encodes a low-complexity protein of unknown function, while *CG30172* was previously described as the insect odorant-binding protein A10, a member of the ejaculatory bulb-specific protein 3 superfamily [31]. Therefore, we decided to focus our study on the protein of unknown function (encoded by *CG9129*), here after named Booster for Crq (Bfc).

## *crq* expression is regulated by Bfc *in vivo*

The first wave of apoptosis in *Drosophila* begins at stage 11 (approximately 7 h after egg laying) of embryogenesis at 25˚C [32]. The macrophages originate from the cephalic mesoderm and spread throughout the embryo. Similar to the pattern of apoptosis, Crq protein is expressed in hemocytes/macrophages from late stage 11 of embryogenesis [15], consistent with its function as an engulfment receptor. We observed that reduced *bfc* levels resulted in defective efferocytosis *in vitro* (**Fig 3A**). To understand whether the relationship between Bfc and *crq* in the context of efferocytosis also existed *in vivo*, we generated a *bfc^{ko}* mutant using CRISPR/Cas9 technique. Significantly, the sequencing results showed early termination of the predicted Bfc protein (**Fig 3B**); additionally, qPCR and western blot analyses confirmed successful establishment of a null allele mutant (**Fig 3C and 3D**). Of note, the *bfc^{ko}* mutants were viable and lacked obvious developmental defects. Next, we collected wild-type and *bfc^{ko}* embryos from stages 7 to 14 and analyzed the *crq* and *bfc* expression by qPCR. As anticipated, the *bfc* and *crq* mRNA levels were upregulated in wild-type embryos as apoptosis occurred at stages 9 and 11 (**S3A and S3B Fig**). On the contrary, the *crq* mRNA levels did not increase in *bfc^{ko}* embryos at stage 11 (**S3C Fig**); however, the levels of Crq protein increased in stage 11 *bfc-null* embryos (**S3D and S3E Fig**). One possible explanation for these discrepant results is that some translated Crq may undergo post-translational modifications in *bfc^{ko}* embryos at stage 11 to become stable. Additionally, while the Crq protein levels increased from stages 12 to 14 in wild-type embryos, *bfc^{ko}* embryos failed to upregulate the Crq protein expression at the corresponding stages (**S3D and S3E Fig**). To further confirm that the upregulation of *crq* and *bfc* was induced by ACs, we also quantified the *crq* and *bfc* mRNA levels in stage 10 to 13 *H99* embryos (in which apoptosis does not occur); importantly, both expression levels were lower in *H99* compared to those in the wild type embryos and did not increase during embryonic development (**S3F Fig**). Altogether, these results suggest that the *crq* expression is impacted by Bfc during AC clearance in *Drosophila* embryogenesis.

In addition to Crq, Drpr and SIMU are the other two phagocytic receptors expressed in *Drosophila* embryonic macrophages as well as in "non-professional" phagocyte glia and the ectoderm [10]. To ascertain whether *bfc* specifically regulates *crq* expression *in vivo*, we measured the parallel expression of *drpr* and *simu*. We observed that the *drpr* mRNA expression was upregulated in S2 cells co-incubated with ACs (**S3G Fig**), and no significant decrease in the *drpr* expression was detected in *Bfc* knockdown S2 cells (**S3H Fig**). Importantly, the decreased expression of Crq in *bfc^{ko}* mutant embryos was consistent with our *in vitro* results; of note, the loss of *bfc* specifically impacted the *crq* expression (and not the *drpr* expression) both in S2 cells and *Drosophila* embryos (**S3H** and **S4A Figs**). In addition, the mRNA levels of *drpr*, *crq*, and *simu* in *srp* mutant embryos showed a decreased expression, which was consistent with the results reported in a previous study (**S4A Fig**)[26]. However, *drpr* and *simu* expression was not decreased in *bfc^{ko}* mutant embryos (**S4A Fig**). Taken together, these results show that *in vivo*, the expression of Bfc, Crq, and Drpr is regulated by the presence of ACs, but only the upregulation of Crq expression is mediated by Bfc.

## Macrophages require Bfc for efficient efferocytosis

Compared with control-treated S2 cells, *bfc* RNAi-treated S2 cells engulfed much fewer ACs, exhibiting a similar efferocytosis defect as *crq* RNAi-treated S2 cells. To address whether Bfc also impacts efferocytosis *in vivo*, first we looked at macrophage development by performing acridine orange (AO) staining, as AO can stain all apoptotic cells engulfed by macrophages in the *Drosophila* embryo [28]. We found clustered AO-stained ACs in wild-type embryos. Importantly, *bfc^{ko}* mutant embryos showed comparable results, suggesting that the Bfc

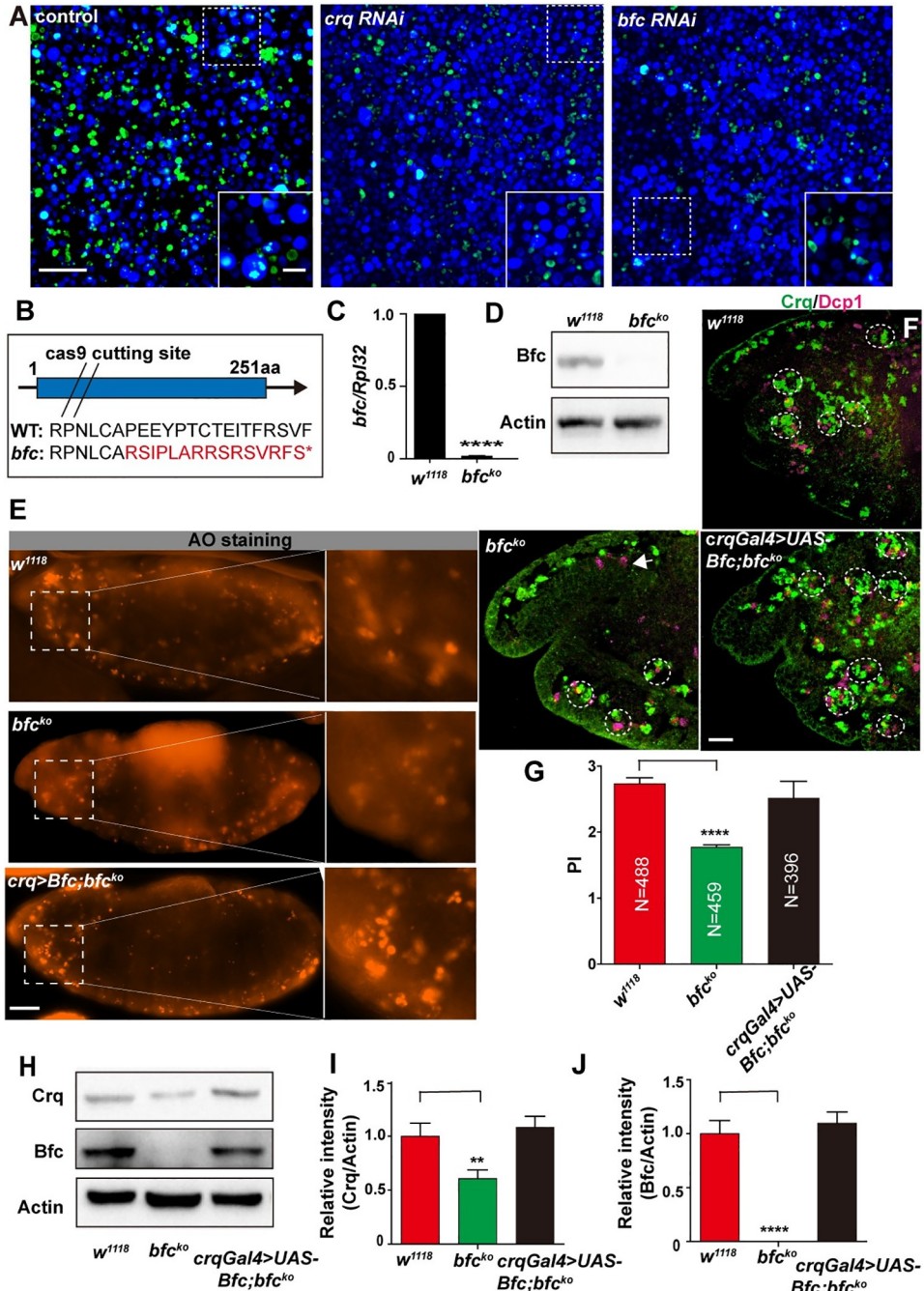

**Fig 3. *bfc* in macrophages and S2 cells is required for efficient efferocytosis. A.** Confocal images showing AC efferocytosis by control-, *crq* RNAi-, and *bfc* RNAi-treated S2 cells (the efferocytosis efficiency is quantified in Fig 2C). Live cells are shown in blue, while cells that engulfed FITC-labeled ACs are shown in green. Scale Bars = 50 μm. **B.** Schematic diagram of the *bfc^ko^* mutant created by CRISPR/Cas9. The exon is denoted as a blue box, and the transcriptional direction is highlighted with an arrow. The double slash represents the Cas9 cutting site. The red marked amino acids represent the mutated sequence. "*" represents the introduced termination codon. **C.** The *bfc* mRNA levels were quantified by qPCR in *w^1118^* and *bfc^ko^* mutant samples. **D.** The Bfc protein levels were also quantified via western blotting in *w^1118^* and *bfc^ko^* mutant samples. Actin was used as the loading control. **E.** AO-stained wild-type, *bfc^ko^* mutant, and *bfc^ko^* mutant re-expressing *UAS-bfc* (under the control of a *crq-Gal4* driver) stage 13 embryos. The insets represent the higher-magnification views of the demarked AO-stained corpses. Scale bar = 20 μm. **F.** Confocal stacks of the head region of stage 13 embryos on the lateral view. Macrophages and apoptotic bodies were stained with anti-Crq (green) and anti-Dcp1 (magenta) antibodies, respectively in wild-type, *bfc^ko^* mutant, and *bfc^ko^*; *crqGal4>UAS-bfc* embryos, which were imaged using the enhanced similarly to show macrophages. Scale bar = 20 μm.

**G.** Graph showing the mean PI ± SEM for each genotype showed in (**F**). **H.** Western blot analysis of the Crq and Bfc protein levels in stage 13 embryos of $w^{1118}$, $bfc^{ko}$ mutant, and $bfc^{ko}$ mutant re-expressing *UAS-bfc* (under the control of a *crq-Gal4* driver). Actin was used as the loading control. **I-J.** Quantification of the Crq and Bfc protein levels in (**H**) after normalization to the Actin levels; n = 3. Statistical analysis was performed using the student's *t-test*(**C**) and one-way ANOVA(**G, I, J**).

knockout did not affect the development of macrophages (**Fig 3E**). In addition, the number of macrophages and ACs in the embryos was quantified using Crq (macrophage marker) and *Drosophila* Dcp1 (AC marker) staining. We found that macrophages developed normally and migrated throughout in stage 13 embryos, and there was no apparent difference in the number of ACs, further supporting the hypothesis that Bfc knockout does not affect macrophage development (**S4B** and **S4C Fig**). To quantify the engulfment of apoptotic bodies by macrophages, we also counted the number of apoptotic particles per macrophage, termed as the "phagocytic index"(PI) [33]. We observed that wild-type macrophages efficiently engulfed ACs, with a PI of 2.74 ± 0.1, while $bfc^{ko}$ mutant macrophages showed reduced AC engulfment, with a lower PI of 1.78 ± 0.04 (35% reduction) (**Figs 3F,3G, S5A and S5D**). This difference was also observed in the context of a *Srp-Gal4,UAS-GFP* marker model; control embryos showed ACs mostly inside GFP-positive macrophages (**S5B Fig**), while in $bfc^{ko}$ mutant embryos, most apoptotic particles (detected in lower abundance) were outside GFP-positive macrophages (**S5C and S5D Fig**). Moreover, we detected decreased expression of Crq in $bfc^{ko}$ mutant embryos, suggesting that the downregulated Crq expression in the absence of *bfc* leads to defective AC clearance (**Fig 3H, 3I and 3J**). Importantly, to prove that the obtained phenotype was indeed dependent on the *bfc* knockout, we generated a transgenic line encoding the full-length cDNA isoform *UAS-bfc* and expressed *bfc* in $bfc^{ko}$ mutant macrophages driven by the macrophage-specific *crq-Gal4* promoter. We observed that the Bfc expression was restored in mutant macrophages and fully rescued the AC clearance defect (PI of 2.52 ± 0.25), as supported by the presence of several apoptotic bodies within the majority of complemented macrophages (**Fig 3F and 3G**). Of note, the restoration of Crq and Bfc expression levels to those quantified in wild-type cells was also observed (**Fig 3H, 3I and 3J**). Finally, as an alternative approach to confirm the role of Bfc in efferocytosis, we used a mutant from Bloomington *Drosophila* Stock Center with a MiMIC insertion [34] in the first exon region of the *bfc* locus, $bfc^{MI02020}$. This insertion produced a potentially strong loss-of-function allele: the *bfc* expression levels were extremely low levels compared to those detected for the wild-type allele (**S6A Fig**). Importantly, homozygous $bfc^{MI02020}$ macrophages in stage 13 embryos showed a significantly impaired phagocytic ability with a PI of 2.0 ± 0.4, and the transgenic *Bfc* expression driven by a macrophage-specific *crq-Gal4* promoter fully rescued the AC clearance defects with a PI of 2.8 ± 0.3, thus supporting the hypothesis that Bfc is required for efferocytosis *in vivo* (**S6B and S6C Fig**). Of note, the restoration of Crq expression in $bfc^{ko}$ macrophages to the levels observed in wild-type cells also completely rescued the AC clearing defect, with a PI of 2.9 ± 0.44 (**S6D and S6E Fig**), highlighting the importance of the cross-talk between Bfc and Crq in efferocytosis. Altogether, our results demonstrate that in embryonic macrophages, Bfc regulates the expression of Crq, and is required for an efficient efferocytosis *in vivo*.

## Bfc physically interacts with the GATA factor Srp

As Bfc was shown to regulate *crq* expression at the transcriptional level, we initially hypothesized that it encodes a transcription factor. However, bioinformatics analysis showed that Bfc is an uncharacterized protein consisting of 251 amino acids with a predicted molecular weight of 29 kD, and without a conserved transcription factor domain. Significantly, protein sequence

alignment showed no homolog of Bfc in vertebrates. Therefore, to clarify the function of Bfc in efferocytosis, next we examined the Bfc localization in embryonic macrophages via immuno-fluorescence staining with anti-Bfc and anti-Crq antibodies. Expectedly, our results revealed Bfc protein localization in macrophages (**Fig 4A**). Interestingly, the subcellular localization analysis revealed that Bfc-GFP fusion proteins were predominantly localized in the nucleus of S2 cells (**Fig 4B**). Of note, anti-Bfc staining also showed that endogenous Bfc was localized in the nucleus of both S2 cells and macrophages (**Fig 4B', 4B" and 4B"'**). Since Bfc regulates the transcriptional expression of *crq*, and localized in the nucleus, we hypothesized that Bfc acts as a co-factor of (a) *crq*-specific transcription factor(s).

Previous studies have demonstrated that several signaling pathways, such as d-JNK, PI3K, and insulin signaling, regulate the expression of the engulfment receptor Drpr [35–37]. In particular, Serpent (*Srp*), a critical transcriptional regulator of macrophage development, is required for apoptotic cell clearance by embryonic macrophages through the regulation of the expression of SIMU, Drpr, and Crq [26]. Therefore, to understand whether Bfc serves as a cofactor of Srp, we used yeast two-hybrid interaction analysis. Importantly, we found that Bfc indeed interacted with Srp in yeast cells (**Fig 4C**). We further confirmed that Bfc interacts with Srp in S2 cells by performing co-immunoprecipitation experiments using Flag-tagged Bfc and HA-tagged Srp expressing cells (**Fig 4D**). Of note, the knockdown of Bfc did not impact Srp localization (**Fig 4E and 4E'**), strongly suggesting that Bfc serves as a cofactor of the Srp-mediated transcriptional regulation.

## Bfc and Srp act together to regulate crq expression in the context of efferocytosis

Our results clearly showed that Bfc physically interacts with Srp. To address whether the Bfc-Srp binding exerted the physiological effects observed *in vivo*, we examined the transcriptional and translational relationship between *bfc* and *srp* expression. To this end, we first detected the Crq transcript and protein levels in $bfc^{ko}$, *srp* mutant [26], and $bfc^{ko}$/*srp* double heterozygous embryos. The results showed that the Crq levels decreased to a similar extent in $bfc^{ko}$, *srp* mutant, and $bfc^{ko}$/*srp* double heterozygous embryos (**Fig 5A, 5B and 5C**). Furthermore, embryos carrying the double heterozygous combination of $bfc^{ko}$ and *srp* (PI = 1.7 ± 0.15) exhibited similar defects to $bfc^{ko}$ (PI = 1.67 ± 0.18) and *srp* (PI = 1.55 ± 0.13) in AC clearance (**Fig 5D and 5E**) as single homozygous mutants, while single heterozygous macrophages showed no clear phenotypic difference from that observed in wild type cells (PI = 3.27 ± 0.2) (**S6E and S6F Fig**). These results imply that Bfc acts together with Srp to upregulate Crq expression to promote an efficient efferocytosis.

As Srp is required for both the Crq expression and AC clearance, we speculated that *bfc* also affects the *crq* levels and efferocytosis in the same pathway. We generated a $crq^{ko}$ mutant using CRISPR/Cas9 technique, and crossed it with the $bfc^{ko}$ mutant to obtain $crq^{ko}$/$bfc^{ko}$ double heterozygous mutants. qPCR and western blot analysis showed that the double heterozygous $crq^{ko}$/$bfc^{ko}$ mutant embryos exhibited comparable decreased Crq levels as $bfc^{ko}$ mutants, and phagocytic defects with reduced AC engulfment(PI = 1.32±0.15) as those observed in $bfc^{ko}$ mutants (**Fig 5F, 5G, 5H, 5I and 5J**). Therefore, we hypothesized that Bfc serves as an Srp cofactor to promote the *crq* transcription and AC clearance during embryonic development.

## Bfc enhances direct binding of GATA factor Srp to the *crq* promoter

The GATA factor Srp regulates the expression of Crq in embryonic macrophages [26]. To dissect how Bfc cooperates with Srp to regulate *crq* transcription, first, we identified the domains in Srp targeted by Bfc. We constructed truncated variants of Srp for Y2H and Co-IP assays

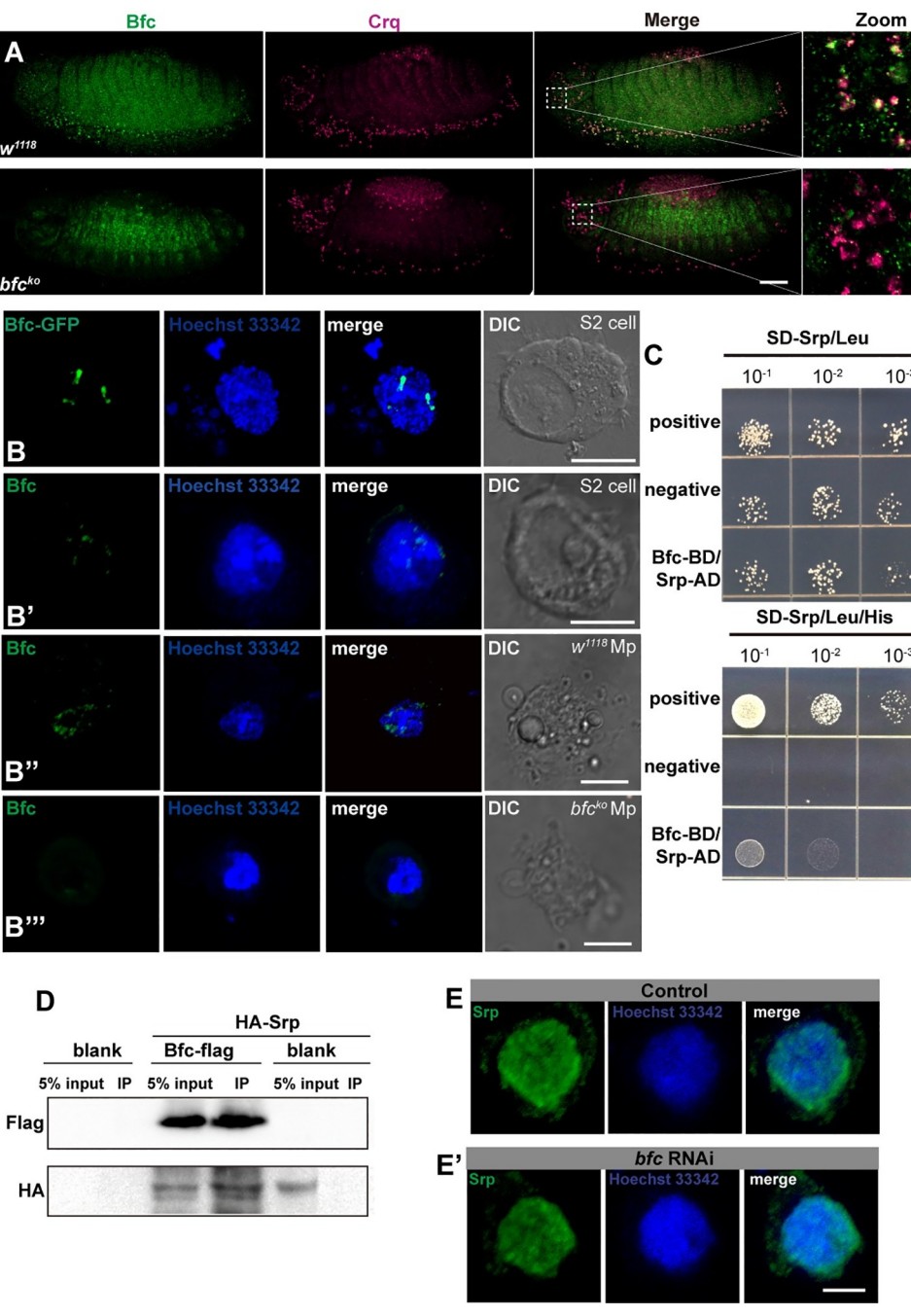

**Fig 4. *bfc* is expressed in embryonic macrophages, and physically interacts with the GATA factor Srp. A.**
Macrophages of stage 13 *w*$^{1118}$ and *bfc*$^{ko}$ embryos were stained with anti-Crq (magenta) and anti-Bfc (green)
antibodies. Scale bar = 20 μm. **B**. Representative fluorescence pictures of the Bfc-GFP expression in S2 cells; **B'-B'''.**
Anti-Bfc (green) stained S2 cells (**B'**); macrophages (showed Mp for short in figure) isolated from *w*$^{1118}$ larvae (**B''**) and
*bfc*$^{ko}$ larvae (**B'''**); the nuclei were stained using Hoechst 33342 (blue). Scale bar = 5 μm. **C.** Yeast two-hybrid assays to
detect the interaction of Bfc (Bait) with Srp (Prey). Different concentrations of the labeled yeast transformants were
grown on SD-Trp-Leu-His plates. **D**. Crude protein extracts from blank S2 cells, transiently transfected S2 cells
expressing Bfc-Flag, and HA-Srp or HA-Srp alone were immunoprecipitated with anti-Flag magnetic beads. WB
detection was performed using anti-Flag and anti-HA antibodies. **E-E'.** Anti-Srp (green) stained control S2 cells as well
as *bfc* RNAi-treated S2 cells; the nuclei were stained using Hoechst 33342 (blue). Scale bar = 5 μm.

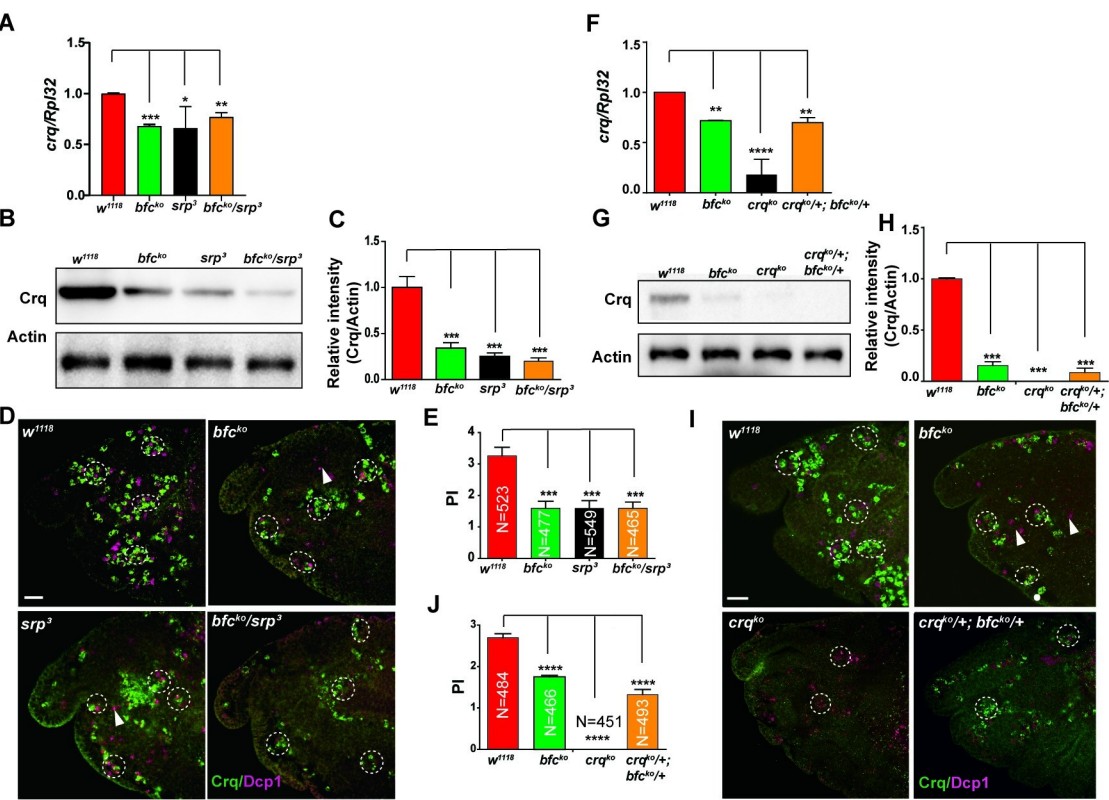

**Fig 5. Bfc interacts with Srp to regulate *crq* expression and efferocytosis. A.** The relative *crq* mRNA levels were quantified by qPCR in macrophages obtained from wild-type, homozygous *bfc^ko^* or *srp^3^*, and double heterozygous *bfc^ko^* and *srp^3^* stage 13 embryos. **B.** Western blot analysis of the Crq protein levels in macrophages of wild-type, homozygous *bfc^ko^* or *srp^3^*, and double heterozygous *bfc^ko^* and *srp^3^* stage 13 embryos. Actin was used as the loading control. **C.** Quantification of the Crq protein levels in (**B**) after normalization to the Actin levels; n = 3. **D.** Macrophages of wild-type, homozygous *bfc^ko^* or *srp^3^*, and double heterozygous *bfc^ko^* and *srp^3^* stage 13 embryos were stained with anti-Crq (green) and anti-Dcp1 (magenta) antibodies. Scale bar = 20 μm. **E.** Graph showing the mean PIs ± SEM for each genotype in (**D**). **F.** The relative *crq* mRNA levels were quantified by qPCR in macrophages of wild-type, homozygous *bfc^ko^* or *crq^ko^*, and double heterozygous *bfc^ko^* and *crq^ko^* stage 13 embryos. **G.** Western blot analysis of the Crq protein levels in macrophages of wild-type, homozygous *bfc^ko^* or *crq^ko^*, and double heterozygous *bfc^ko^* and *crq^ko^* stage 13 embryos. Actin was used as the loading control. **H.** Quantification of the Crq protein levels in (**G**) after normalization to the Actin levels; n = 3. **I.** Macrophages of wild-type, homozygous *bfc^ko^* or *crq^ko^*, and double heterozygous *bfc^ko^* and *crq^ko^* stage 13 embryos were stained with anti-Crq (green) and anti-Dcp1 (magenta) antibodies. Scale bar = 20 μm. **J.** Graph showing the mean PIs ± SEM for each genotype in (**I**). Statistical significance was assessed using the one-way ANOVA.

(**S7A Fig**). These assays revealed that the zinc finger (ZnF) GATA domain of Srp is important for Bfc binding (**S7B** and **S7C Fig**). Notably, the ZnF GATA domain found in Srp has been described as a DNA-binding domain; the GATA factor was reported to have a C4 type zinc finger for binding to DNA (**S7D Fig**) [38]. Importantly, the mutant *Srp^ZnF^* (Cys to Arg) showed impaired interaction between Srp and Bfc, indicating that Bfc interacts with a non-DNA binding region in the Srp ZnF domain (**S7E Fig**).

To detect the physical interactions between regulatory transcription factor proteins and *crq* at the genome level, we performed yeast one-hybrid (Y1H) assays [39] between *Srp* and the *crq* promoter locus. The results showed that Srp-AD interacted only with the *crq* promoter-3 (-401~800 bp upstream of *crq*) (**Fig 6A and 6B**). Importantly, sequencing analysis of the -1600 bp *crq* promoter element led to the discovery of one potential Srp-binding site [(A/T)GATA (A/G)], 494 bp upstream of *crq*, suggesting that *Srp* might regulate the transcription of *crq* via direct binding to its promoter. As expected, in the context of the mutant Srp^ZnF^ (from Cys to Arg), binding to the *crq* promoter was not detected (**S7F Fig**). To confirm that the GATA

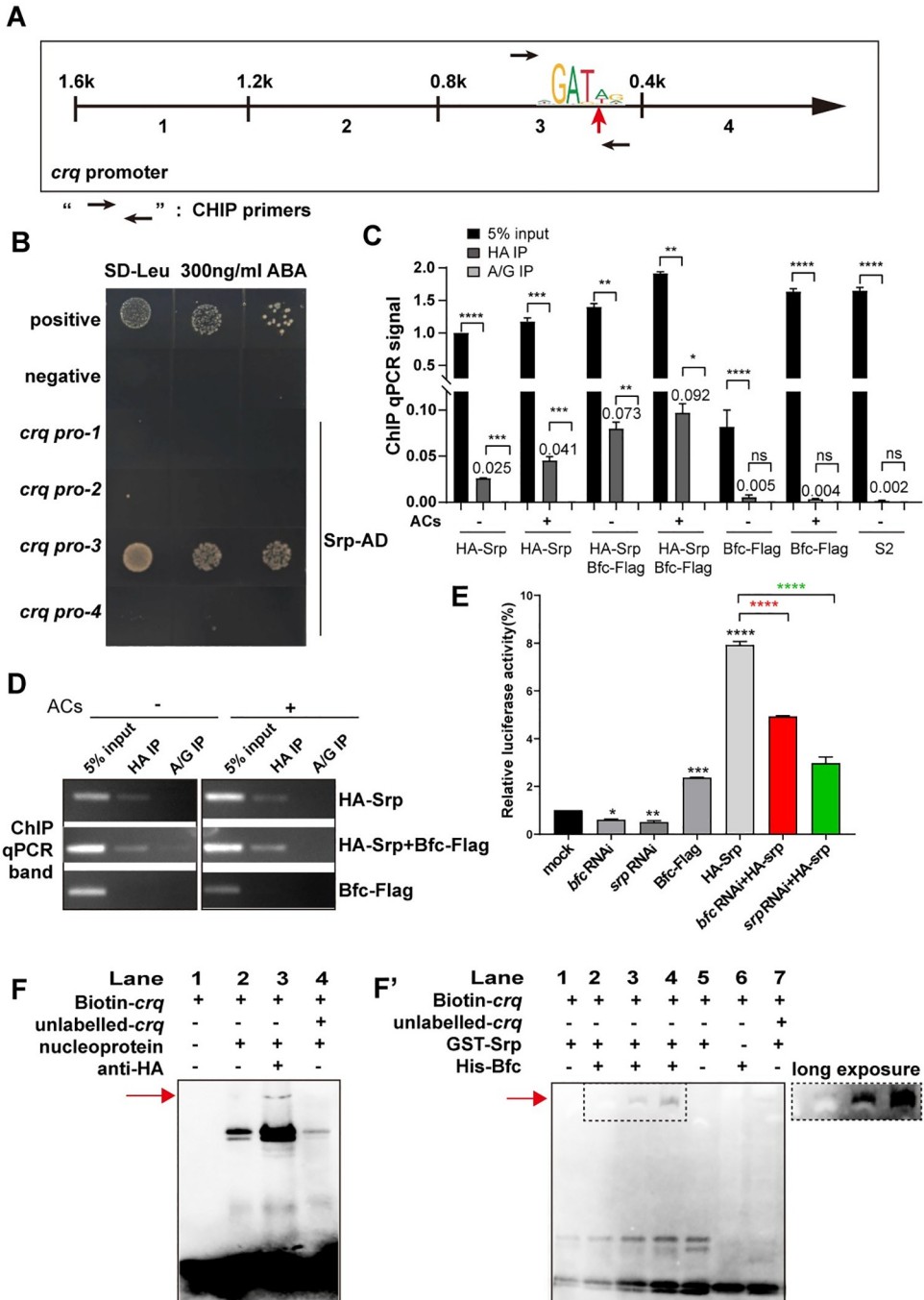

**Fig 6. Srp binds to the GATA motif in the *crq* promoter. A.** Schematic representation of the *crq* promoter region; the red arrowhead indicates the predicted Srp binding site, and the black arrows indicate the primers used for the ChIP-qPCR. **B.** Yeast One-hybrid assays with yeast strains containing different prey (Srp-AD)—bait (*crq* promoter plasmids) combinations. **C.** The crosslinked chromatin of S2 cells expressing HA-Srp, HA-Srp+Bfc-Flag, or Bfc-Flag, incubated or not with ACs was immunoprecipitated with either anti-HA beads or control protein A/G. The ChIP-qPCR efficiency (compared to 5% input) is shown. Protein A/G was used as the negative control. **D.** The gel image shows the input DNA or immunoprecipitated DNA bands of the ChIP-qPCR in (**C**). **E.** Effect of *bfc* and *srp* RNAi treatments on the luciferase activity of *crq* promoter reporter constructs. All the samples were tested after the addition of ACs. The mean activity with the standard deviation of three independent transfections is represented. Statistical significance was assessed using the one-way ANOVA. **F.** Nuclear proteins extracted from HA-Srp transfected S2 cells binding to biotin-labeled *crq* promoter probe and EMSA supershift with and without anti-HA antibodies. The red arrows point to supershifted bands. **F'.** Purified GST-Srp protein binding to biotin-labeled *crq* probe and supershift with purified Bfc-6XHis.

sequence in the *crq* promoter is required for *crq* expression, we mutated the GATA sequence in the Srp-binding site and integrated this construct, or the *crq*$^M$-Gal4 construct (expression of Crq-Flag), as a control into the same genomic location. Our results showed that the ability of *crq*$^M$-*Gal4* to activate *crq* transcription was much weaker than that of *crq-Gal4* (**S8A, S8C, S8D and S8E Fig**); of note, the deficiency in *bfc* also led to the insufficient activation of *crq* (**S8A, S8B, S8D, S8E and S8F Fig**). Moreover, we expressed the GFP protein drived by the mutated *crq* promoter to test its unspecific transcriptional activation, which showed *crq*$^M$-*Gal4* still had a relatively weak ability to activate *UAS-GFP*. Besides, the GFP signal in *crq*$^M$-*Gal4*/ *UAS-GFP*; *bfc*$^{ko}$ was as weak as that in the context of *crq*$^M$-*Gal4*/ *UAS-GFP* (**S9A, S9B and S9C Fig**), which indicates that Bfc regulates *crq* transcription in a GATA-dependent manner. Moreover, ChIP qPCR assays showed that the amplification of the *crq* promoter region containing GATA in HA-Srp-transfected S2 cells (immunoprecipitated with anti-HA antibodies) was 25-fold higher than that obtained with control IgG antibodies (**Fig 6C and 6D**). Remarkably, the addition of ACs further increased the abundance of *crq* region co-immunoprecipitated with Srp (up to 40-fold *versus* the control IgG condition), supporting the idea that ACs induce the Srp-regulated transcription of *crq*. Furthermore, when we expressed Bfc in HA-Srp-transfected S2 cells, the *crq* region was significantly enriched (3-fold *versus* HA-Srp single-transfected S2 cells) (**Fig 6C and 6D**), while the overexpression of Bfc had no obvious influence on the Srp protein levels regardless of the presence of ACs (**S9D and S9E Fig**).

To examine the requirement of Bfc and Srp for *crq* promoter activity during AC clearance, we conducted transient luciferase expression assays with the wild-type *crq* gene promoter-luciferase reporter gene; S2 cells were treated with *bfc* dsRNA or *srp* dsRNA after the addition of ACs. Knockdown of *bfc* or *srp* reduced the *crq* gene promoter activity by 39% and 50%, respectively (**Fig 6E**). On the other hand, Srp overexpression led to a 7.8-fold increase in the *crq* gene promoter activity; however, the knockdown of *bfc* in Srp-overexpressing S2 cells lowered the activity by 4.9-fold (**Fig 6E**). These results indicate that Bfc and Srp are required for *crq* promoter activity. To further investigate the role of Bfc in *crq* promoter activity, we performed electrophoretic mobility shift assay (EMSA) assays using Srp protein *in vivo* and *in vitro*. HA-Srp-transfected S2 nuclei formed a protein-DNA complex with biotin-labeled *crq* promoter DNA, whose migration was 'shifted' by the addition of anti-HA antibodies, indicating that Srp binds to the GATA sequence specifically (**Fig 6F**). Consistently, the addition of purified Bfc protein (1 μg) to the purified Srp-DNA complex also showed a super-shifted pattern compared to that of Srp-DNA alone (**Fig 6F', Lane 2**). Of note, when we added increased amount of Bfc (2 μg and 4 μg) to the purified Srp-DNA complex, a more significant supershift was observed (**Fig 6F', Lane 3–4**). As PI3K signaling and Stat92e reportedly activate the transcription of *drpr* [36], and Srp is required for the expression of Crq and Drpr [26], we further investigated whether Srp directly regulates the expression of phagocytic receptors. The Y1H results showed that only Srp, and neither Bfc nor Stat92e, bound directly to the *crq* promoter. However, we did not detect the binding of Srp to the *drpr* promoter (**S10 Fig**). Overall, these findings suggest that Srp directly binds to the *crq* promoter in conjunction with Bfc to regulate the Crq levels; thus, these proteins function together in regulating efferocytosis.

## Discussion

In mammals, ACs are recognized by CD36, one of the several phagocyte cell surface receptors, with the AC surface molecules serving as cognate "eat-me" signals/ligands [40,41]. ACs also secrete molecules that attract distant phagocytes and modulate the immune response or phagocytic receptor activity. However, the mechanisms underlying this effect remain unclear.

Crq is a CD36-related scavenger receptor in *Drosophila* and is expressed immediately after the onset of apoptosis in embryonic macrophages [8]. The expression of Crq is regulated by the extent of apoptosis, although the regulatory mechanisms by which ACs control the expression of Crq and subsequently induce phagocytosis in embryonic macrophages have not been described.

Here, we reveal a novel protein, Bfc (Booster for *Crq*), that plays a key role in efferocytosis via specifically regulating the expression of *crq* in a manner dependent on the extent of apoptosis. Bfc interacts with the zinc finger domain of the transcription factor Srp as a cofactor to enhance the binding of Srp to the *crq* promoter, leading to the upregulation of *crq* expression and the consequent induction of efferocytosis in *Drosophila melanogaster*. Importantly, our data reveal the molecular mechanisms by which ACs affect Crq expression, as well as how the phagocytic ability of embryonic macrophages is boosted in the presence of excessive apoptosis.

We found that in S2 cells, the ACs induced the transcriptional upregulation of *crq* (**Fig 1A**). *In vivo*, the macrophages developed as early as the first wave of developmentally programmed apoptosis began at embryogenic stage 11, when the expression of *crq* was activated and subsequently became widespread throughout the embryo (**S3A, S3B, S3C and S3D Fig**). Importantly, these results are similar to the regulatory mechanisms associated with the expression of other phagocytic receptors, such as Drpr and integrin [42,43]. For instance, studies showed that AC engulfment rapidly triggers an intracellular calcium burst followed by increased levels of *drpr* transcripts in *Drosophila* macrophages [16]; similarly, Draper and integrins become apically enriched soon after the engulfment of apoptotic debris in epithelial follicle cells [44].

We found that the expression of *crq* was elevated early after the co-culture of ACs and S2 cells, but gradually decreased to the basal levels as efferocytosis continued, suggesting that the regulation of AC clearance and *crq* expression follow a similar pattern. We demonstrated that most AC samples added to live S2 cells were composed of apoptotic cells rather than necrotic cells. However, we cannot eliminate the upregulated expression of genes in response to the presence of a few necrotic cells. Indeed, based on transcriptome analysis, we identified 12 genes that are required for AC clearance, which was confirmed by subsequent efferocytosis assays using their individual knockdown in S2 cells. Interestingly, among the 12 genes, two were related to innate immunity. *CecA1*, regulated at the transcriptional level encodes an antibacterial peptide [45], as well as a secreted protein that mediates the activation of the Toll pathway during bacterial infection [46]. This result may contradict the discreet nature of the apoptotic process [47,48]. However, ATPs released by bacteria are known to mediate inflammation [49], and the toll-like receptor 4 (TLR4) is activated by ACs to promote dendritic cell maturation and innate immunity in human monocyte-derived dendritic cells [50]. These results indicate that innate immune pathways are activated in the presence of ACs, and may contribute to their recognition or clearance in *Drosophila*.

Among these 12 genes, *CG9129 (bfc)* and *CG30172* regulated the expression of *crq* and hence, efferocytosis. Further studies must be performed to elucidate the role of *CG30172* in efferocytosis. On the other hand, we clearly dissected the role played by *bfc* in efferocytosis as well as the underlying mechanism. Using several different experimental approaches, we demonstrated that *bfc* regulates *crq* expression in response to excessive apoptosis. First, *bfc* RNAi treatment decreased the *crq* expression levels in S2 cells exposed to ACs, but not in the absence of ACs. Second, the increase in *crq* transcription was proportional to the extent of apoptosis in embryos, which was blocked by the loss of *bfc*. Notably, other phagocytic receptors have been reported to be activated by dying cells. The integrin heterodimer αPS3/βPS can be enriched in epithelial follicle cells after the engulfment of dying germline cells [42]. In addition, Drpr expression increases in follicle and glial cells, which activates the downstream JNK signaling during the clearance of apoptotic germline cells and neurons, respectively[35,43]. Collectively,

the available scientific literature suggests that the expression of phagocytic receptors can be stimulated by the presence of excessive ACs to improve the phagocytic activity of macrophages or epithelial cells in different tissues.

Bioinformatics analysis of the conserved domains and gene structure indicated that Bfc does not *likely* function directly as a *transcription factor*. Here, we identified Srp as a Bfc interaction partner using yeast two-hybrid and Co-IP analyses. Shlyakhover *et al*. [26] reported that Srp is required for the expression of SIMU, Drpr, and Crq receptors in embryonic macrophages; however, our results demonstrated that *bfc* only affects the expression of *crq* expression through interaction with Srp, with no impact on the expression of several other genes. A plausible hypothesis for this phenotype is that Bfc assistance for Srp binding to the promoters of *simu* and *drpr*, may have limited effects. Thus, our results suggest that Bfc may regulate the Crq expression levels in the first wave of AC recognition via binding to Srp, whereas other regulatory factors participate in the Srp-mediated regulation of Drpr and SIMU.

We found that Srp directly binds to the DNA consensus sequence GATA [51] of the *crq* promoter via its highly conserved Cys-X2-Cys-X17-Cys-X2-Cys zinc finger binding domain (C4 motif) (**Figs 6 and S7D**). Meanwhile, Srp also interacts with Bfc through its zinc finger domain; curiously, while the mutation of the C4 motif did not affect the latter interaction, it completely blocked the former (**S7 Fig**). Importantly, we also showed that mutation in the GATA site abolished the expression of the *crq* in *Drosophila* embryo macrophages (**S8 Fig**). As a potential Srp cofactor, Bfc increased the ability of Srp to bind to the *crq* promoter, while *bfc* knockdown inhibited the *crq* transcriptional activity (**Fig 6E and 6F**). Ush (homolog of FOG-2 in *Drosophila*), a cofactor of GATA transcriptional factors [52], can bind Srp and limit crystal cell production during *Drosophila* blood cell development [53]. Interestingly, genetic studies have demonstrated that Ush acts with Srp to maintain the pluripotency of hemocyte progenitors and suppresses their differentiation [54,55]. Ush was reported to repress *crq* expression by interacting with the isoform of Srp, SrpNC(with two GATA zinc finger) while the other isoform of Srp, SrpC(with one GATA zinc finger) induced *crq* expression [25], which may indicate Bfc and Ush act on different isoforms of Srp to regulate *crq* expression by opposite mechanisms.

Although our results elucidate several factors that contribute to efferocytosis in *Drosophila* embryos, some mechanistic details remain unresolved; for instance, how ACs induce Bfc-mediated regulation of *crq* expression in macrophages remains unclear. We found that Bfc regulates Crq expression and efferocytosis, but not macrophage development (**Fig 3E**). Moreover, we found that Bfc-mediated activation of *crq* transcription and Crq accumulation leads to positive feedback to promote increased Bfc expression, which is required for engulfment (**S2 Fig**). As expected, the upregulation of Bfc expression occurred earlier than that of *crq* in S2 cells after incubation with ACs (**S3F Fig**). Therefore, further studies are required to elucidate the upstream signals in the context of the *crq*-mediated regulation of *bfc* expression. As previous studies have shown that Crq is required for phagosome maturation during the clearance of neuronal debris by epithelial cells and bacterial clearance [13,56], further studies should be conducted to determine whether Bfc is involved in the clearance of neuronal debris.

This study is not without limitations. For instance, other potential regulators of efferocytosis, whose expression is not affected by ACs could not be detected in our study. In mammals, CD36 is involved in the clearance of ACs [57] and regulates the host inflammatory response [58]. As a CD36 family homolog, Crq promotes the clearance of ACs and bacterial uptake via efferocytosis [56]. Researchers have reported that the GATA factor Srp is required for Crq expression [26]; here we confirmed this finding and showed that Srp directly binds to the *crq* promoter via its GATA binding site, which is enhanced by Bfc. However, no apparent Bfc homologs exist in vertebrates, and whether GATA factors regulate the CD36 family in a

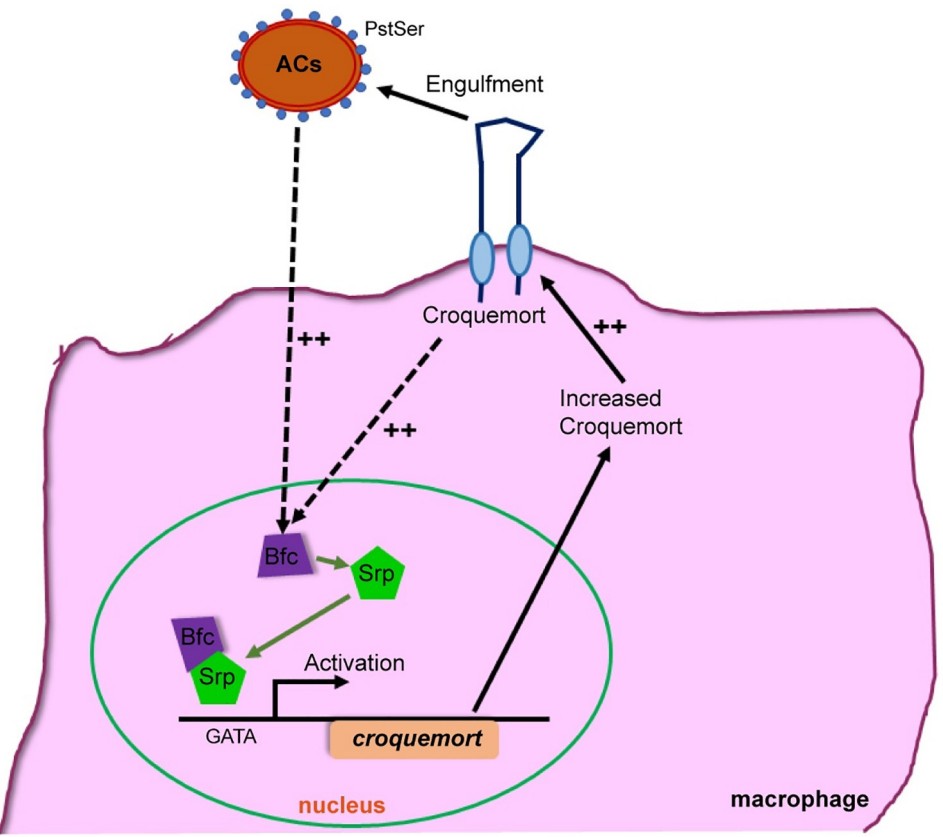

**Fig 7. The AC-induced Crq expression is regulated by the Bfc-Srp pathway.** The basal Bfc expression remains at relatively low levels in the absence of ACs. However, in the presence of ACs, during efferocytosis, *bfc* expression is transcriptionally upregulated earlier than *crq*, and Bfc interacts with the GATA factor Srp to facilitate latter's binding to the *crq* promoter. In turn, increased Crq levels likely enhance the ability of macrophages to engulf ACs and lead to the upregulated expression of Bfc via a positive feedback loop.

mechanism similar to that in flies remains unclear. Nevertheless, we predict that one or more functional homologs of Bfc may exist in mammals and are likely involved in apoptotic cell clearance. Unraveling them as well as determining whether and how *bfc* participates in eliminating pathogens and innate immunity is essential.

In summary, we showed that the expression of the engulfment receptor Crq is transcriptionally regulated by the presence of ACs, via Srp, and its newly identified cofactor, Bfc (Fig 7). Altogether, our findings imply that macrophages adopt a precise mechanism to increase the expression of engulfment receptors to boost their phagocytic activity, in the presence of excessive ACs. We anticipate a similar role and mechanism in the context of mammalian engulfment receptors in response to excessive ACs. Therefore, the findings of this study have significant implications for a wide range of human diseases, including those associated with aberrant apoptotic cell death and efferocytosis, such as tumor progression, neurodegenerative disorders, and other severe inflammatory conditions.

## Materials and methods

### Ethics statement

All experiments were performed in accordance with the approved guidelines. The animal experiments were performed in accordance with the Chinese Guidelines for the Care and Use

of Laboratory Animals and approved by the Animal Care and Use Committee of the Shaanxi Normal University (Xi'an, China).

## S2 cell culture and transfection

*Drosophila* S2 cells were cultured in Sf-900II SFM (Gibco) containing 50 units/ml penicillin and 50 μg/ml streptomycin at 25˚C. Approximately $2\times10^5$ cells were plated 24 h before transfection; 2 μg DNA were transfected into S2 cells via DDAB-mediated transfection [59]. Transfected cells were harvested after 3 days.

## Total RNA extraction and qPCR

Total RNA was extracted from S2 cells and flies using the TRIzol reagent (Invitrogen #15596026) and the Direct-zol RNA miniprep kit (Zymoresearch #R2052). RNA was reverse-transcribed using the Transcriptor First Strand cDNA Synthesis Kit (Roche #04896866001), and qPCR was performed using the SYBR Mix (Roche # 04707516001) in an ABI Step One Plus instrument. Data represent the ratio or relative ratio of mRNA levels of the target gene (such as *crq*) to those of an internal reference gene (*Rpl32*). The qPCR primers used in this study are listed in **S2 Table**. All experiments were performed in triplicate with mRNA samples extracted from at least three biological replicates. Relative gene expression was quantified using the comparative method (ΔΔCT) [60,61].

## Preparation of apoptotic cells and efferocytosis assays using S2 cells

S2 cells were treated with 0.25 μg/ml actinomycin D (Sigma) in culture medium for 18 h to induce apoptosis, washed three times with PBS, fixed with 8% paraformaldehyde for 10 min at room temperature (RT), and resuspended in SFM at a concentration of $10^7$ cells/ml. For mRNA or protein expression analyses, apoptotic cells were directly added to live S2 cells at a ratio of 5:1. For the induction of efferocytosis *in vitro*, 1 ml ACs were labeled with 5 μg FITC-isomer suspension (Invitrogen #F-1906) for 1 h at RT in the dark. The cells were then washed twice with PBS, resuspended in 1 ml SFM, added to live S2 cells, and incubated for 6 h. Cells were counterstained with 10 μM CellTracker Blue CMAC Dye (Invitrogen #C2111) for 1 h, quenched with 0.4% trypan blue (Sigma), and observed under a confocal microscope (Nikon, C2). The acquired images were processed with the Adobe Photoshop CS6 and ImageJ software. As reported in previous studies [28,33], the ratio of double-labeled live cells (cell tracker blue and FITC-green) *versus* all live cells (cell tracker blue) was calculated to determine the percentage of efferocytosis.

## RNA-seq assay

Total RNA of S2 cells was prepared as described above, and analyzed by the Annoroad Corporation. The libraries were constructed and sequenced using a HiSeq 2500 system (Illumina), which generated ~21 million read pairs per sample. Raw reads containing adapter, poly-N, and low-quality reads were filtered, and the effective data were mapped to the *Drosophila melanogaster* reference genome using TopHat (version 2.0.12)[62]. After excluding rRNA or tRNA sequences, we estimated the abundance of transcripts using fragments per kilobase per million mapped fragments (RPKM) [63]. The *P* values were adjusted using the Benjamini and Hochberg method [64]. Corrected *P* values < 0.05, and |log FC| ≥0.4, were set as the threshold for the significant difference in expression. The differentially expressed genes after filtering the RNA-seq data are denoted in the supplemental data.

All RNAseq data files are available from the NCBI SRA database (accession number PRJNA733098). The URLs is https://www.ncbi.nlm.nih.gov/Traces/study/?acc= PRJNA733098.

All other data is available within the paper and its Supporting Information files.

## Protein extraction and western blot analysis

Protein samples were lysed in 50 mM Tris-Cl [pH 7.4], 1% TritonX-100, 0.15 M NaCl, 1 mM EDTA with protease inhibitor. Lysates were centrifuged at 20,000 g for 30 min, and the supernatants were mixed with 5 × protein loading buffer for western blotting. For the embryo samples used for western blotting, we collected embryos laid on a 60 mm×60 mm yeasted apple juice agar plate, which were produced by 100–200 flies for 2 h at 25˚C; the embryonic stages were determined according by Campos-Ortega and Hartenstein (1985).

The antibodies were used at the following dilutions: mouse anti-Crq (1:500), mouse anti-Bfc (1:500) (prepared in our lab), mouse anti-actin (CST), 1:1,000, mouse anti-Flag (Sigma) 1:1000, and mouse anti-HA (CST) 1:1000. Anti-rabbit or anti-mouse horseradish peroxidase (HRP)-coupled secondary antibodies (Jackson ImmunoResearch Laboratories) were used at a 1:10,000 dilution; ECL detection was performed following the supplier's protocol (Pierce).

## RNA interference treatment of S2 cells

Primers for the cloning of dsRNA were designed in *Drosophila* cell-based RNAi screen websites https://fgr.hms.harvard.edu/fly-cell-based-rnai; the T7 promoter sequence TAATAC-GACTCACTATAGGG was added to the 5′ end of each primer, and the High Yield RNA Synthesis Kit (NEB) was used for dsRNA synthesis using the corresponding PCR products as templates. The dsRNA primers used in this study are listed in **S3 Table**. For specific RNAi treatments, 500 μl S2 cells were treated with 5 μg dsRNA for 36 h and then FITC-labeled ACs were added and incubated at 25˚C for 12 h to allow efficient efferocytosis. Subsequently, cells were collected for qPCR analysis and microscopic observation.

## Fly strains and constructs

The following fly strains were used in this study: $w^{1118}$, $bfc^{MI02020}$ (#34268, Bloomington), $srp^3$/ TM3 (#2485, Bloomington), *UAS-Bfc* (Chr II, this study), *UAS-GFP*(#35786, Chr III, Bloomington), *Crq-Gal4* (Chr II, this study), *Crq proM-Gal4* (Chr II, this study), *vasa Cas9* (#51323, Bloomington), *bfc sgRNA* (Chr II, this study), *crq sgRNA*(Chr III, this study), $bfc^{ko}$ (*bfc* mutant obtained using CRISPR/Cas9, this study), and $crq^{ko}$ (*crq* mutant obtained using CRISPR/Cas9, this study).

All flies were generated by Unihuaii Inc., unless otherwise specified. The *crq-Gal4* and *crq proM-Gal4* constructs were generated by PCR amplification of the 4000 bp fragment upstream of the *crq* gene (according to the reported *crq-Gal4* sequence[65]) using the following primers: forward cctttcgtcttcaagaattcTGCGCCAGTTATCGACAGCT, reverse gggatcccggatctggtac-cAATAGCCAAGGTGGGAATAATC; in the *crq proM-Gal4* construct, GATA$^{494}$ was mutated to CAGA$^{494}$. The PCR product was directionally cloned into the Invitrogen pENTR/D-TOPO vector (Invitrogen #K240020), and then cloned into the pBGUw destination vector [15] using the Invitrogen Gateway LR Clonase Enzyme kit (Invitrogen #11791020). The construct was sequence verified, and transgenic flies were generated by Unihuaii Inc. using the PhiC31 targeted integration system.

The *UAS-Bfc* and *UAS-CrqFlag* constructs were constructed by PCR amplification of the genome using the following primers: Bfc forward caccATGAGAATGGACAAGTCGGA, Bfc reverse CTATGTCTTCTTGGCTGGCA; Crq forward caccATGTGCTGCAAGTGCTGCGG;

Crq reverse CTACTTGTCATCGTCATCCTTGT. The same method was used to clone the PCR product into the pENTR/D-TOPO vector, followed by cloning in the pPWG destination vector. The construct was sequence verified, and transgenic flies were generated by Unihuaii Inc. using the P-element system. The *bfc* and *crq* sgRNA constructs were constructed using *bfc* and *crq* sgRNA oligos obtained from http://targetfinder.flycrispr.neuro.brown.edu/. The *bfc* and *crq* sgRNA constructs were generated using the following primers: Bfc sgRNA forward gtcgTCCTCCGGAGCGCAGAGGTT, Bfc sgRNA reverse aaacAACCTCTGCGCTCCGGAG GA; Crq sgRNA forward gtcgAACTTTATGCTGAACCATGA, Crq sgRNA reverse aaacTCA TGGTTCAGCATAAAGTT. The constructs were verified by sequencing, and transgenic flies were generated by Unihuaii Inc. using the PhiC31 targeted integration system. The sgRNA transgenic flies were crossed with the vasa Cas9 strain to generate the Cas9-editing mutants, as verified by qPCR and sequencing.

### Generation of anti-Bfc, -Crq and -Srp antibodies

The anti-Crq rabbit antibodies were a kind gift from Nathalie et al. [8]. Anti-Crq, anti-Bfc, and anti-Srp antibodies were raised via immunization of mice (BALB/c, 6–8-week-old), with the antigens including the amino acid positions 161–491, 1–250, and 911–1264, respectively; all recombinant proteins were fused to a 6×His tag and expressed in *Escherichia coli*. The recombinant DNA constructs used in this study are listed in **S4 Table.** All proteins were expressed in the form of inclusion bodies, resolved by SDS-PAGE, and stained with 0.25 M potassium chloride solution. The target protein bands were collected, and the proteins were dialyzed via horizontal electrophoresis, desalted, and concentrated to immunize the mice.

### Immunostaining

Immunostaining was performed as described previously [26]. Briefly, stage 13 embryos were fixed and stained with different primary antibodies. Rabbit or mouse fluorescein-coupled and Cy5-coupled antibodies (Jackson Laboratories) were used as the secondary antibodies. The stained embryos were mounted in Vectashield medium (Vector Laboratories) and observed under a confocal microscope (Nikon, C2). The antibodies used in this study were rabbit anti-CRQ (1:400), mouse anti-Bfc (1:400), mouse anti-GFP (CST; 1:500), mouse anti-Flag (Sigma; 1:500), rabbit anti-HA (CST; 1:500), and rabbit anti-Dcp1 (CST, 1:500). The goat anti-rabbit Alexa Fluor Plus 594-coupled or anti-mouse Alexa Fluor Plus 488-coupled secondary antibodies (Jackson ImmunoResearch Laboratories) were used at a 1:1000 dilution.

### Acridine Orange staining of *Drosophila* embryos

The AO staining was performed as described previously [66]. Briefly, embryos were collected at stage 13 and dechorionated in fresh 50% bleach until the dorsal appendages were no longer visible. The embryos were then washed with distilled $H_2O$ to remove the bleach and transferred into 1 ml heptane and 1 ml 2.5 μg/ml AO (Molecular Probes) in 0.1 M phosphate buffer, pH 7. Remove the embryos from the interface and dropped them onto a glass slide, and followed by placing the coverslip. Staining was observed in FITC channels.

### Determination of the phagocytic index (PI)

PI was calculated as the average number of apoptotic particles in each macrophage [33]. In general, five embryos per genotype and four parts (head, dorsal side, ventral side, and tail) of each embryo were acquired as confocal image stacks; only the head pictures are shown

[8,15,66]. Confocal imaging was performed on a Nikon C2 confocal microscope. The standard error of the mean (SEM) was obtained from the calculated PIs.

## Yeast two-hybrid assay

Each relevant gene was cloned into the pGADT7 or pGBDT7 plasmid (Clontech). The plasmids were transformed into AH109 yeast strain in pairs. The transformed yeast was cultured on SD–Trp/-Leu double-dropout solid culture medium. After PCR testing, individual positive yeast colonies were picked into a new culture medium and incubated overnight. Colonies were then diluted to $10^{-1}$, $10^{-2}$ and $10^{-3}$ cfu, and added into SD–Trp/-Leu/-His triple dropout solid culture medium. Images of yeast cultures were obtained after 3 d culture at 30˚C.

## Co-Immunoprecipitation (Co-IP)

Plasmids encoding the differently expressed tagged proteins were co-transfected into S2 cells as previously described; immunoprecipitation was carried out at 4˚C. Briefly, transfected S2 cells were lysed in 50 mM Tris-Cl [pH 7.4], 1% TritonX-100, 0.15 M NaCl, 1 mM EDTA with protease inhibitor followed by centrifugation. The supernatants were incubated with anti-Flag M2 magnetic beads (Sigma#M8823) for 2 h at 4˚C. The beads were washed three times with TBS (50 mM Tris, 150 mM NaCl), boiled in 2×SDS loading buffer for 5 min, the supernatants were detected by western blotting.

## Yeast one-hybrid assay

The DNA bait sequence was cloned into the pAbAi vector, for the subsequent integration of the pBait-AbAi plasmid into the Y1HGold yeast genome and the creation of a bait/reporter strain. The background AbA' expression of the Y1HGold bait strain was tested by plating 10 μl yeast on SD/-Ura media supplemented with Aureobasidin A(AbAi) (100–1000 ng/ml); the minimal concentration of AbA, or a concentration that was slightly higher than the one that completely suppresses the growth of the bait strain was used for the screening interacting proteins. The cDNA of indicated transcriptional factor was cloned into pGADT7, transformed into the Y1HGold bait strain, and 10 μl of $10^{-1}$, $10^{-2}$ and $10^{-3}$ dilutions were spotted onto selective medium containing 300 ng/ml ABA and no histidine (-His), followed by 3 d incubation at 30˚C. Positive and negative controls were included for all experiments.

## ChIP assay

10×cell lysis buffer (50 mM HEPES, 50 mM NaCl, 10 mM EDTA, 5 mM EGTA), and Triton X-100 were directly added to a final concentration 1× and 0.5%, respectively to $5 \times 10^7$ exponentially growing cells in 10 ml growth media. The protein-DNA bonds were cross-linked by the addition of 1% formaldehyde, and cross-linking was stopped by adding and incubating 100 mM glycine for 15 min. After centrifugation, the cell pellet was suspended in 200 μl sonic buffer (10 mM Tris pH 8.0, 1 mM EDTA, 0.5 mM EGTA), and sonicated (on ice) to break the DNA into fragments. The lysates were centrifuged for 10 min at $16,000 \times g$, the supernatant was diluted in 800 μL dilution buffer (10 mM Tris, pH 8.0, 1 mM EDTA, 1% Triton X-100, 0.1% sodium deoxycholate, 0.1% SDS, 140 mM NaCl), and incubated with HA magnetic beads (Pierce # 88836) or protein A/G magnetic beads (Pierce # 88802) overnight at 4˚C on a rotating device. The beads were washed five times with dilution buffer. After the last wash, 50 μL elution buffer (1% SDS, 100 mM NaHCO$_3$) was added to elute the immunoprecipitated DNA. The protein-DNA crosslinks were reversed by heating at 65˚C for 4 h. After deproteinization

with proteinase K, DNA was recovered by phenol-chloroform extraction and ethanol precipitation. Finally, 2–10 μL of the DNA per sample was subjected to qPCR.

## Luciferase reporter assays

Approximately $2 \times 10^5$ S2 cells were plated in 24-well dishes 24 h before plasmid transfection; 500 ng pGL3-*crq*Pro plasmid and 1 ng pAct5C-seapanzy as an internal control were co-transfected into cells using DDAB-mediated transfection. After 48 h, S2 cells were harvested, and the luciferase activity was determined according to the manufacturer's instructions (Promega#E1960). Normalization to the *Renilla* luciferase activity was performed. Transfections were performed three times independently. For target gene-RNAi experiments, 5 μg/well dsRNA was added 24 h before plasmid transfection.

## Electrophoretic mobility shift assay

The EMSA has been used extensively to study DNA-protein interactions [67]. Biotin 5' end-labeled DNA targets were prepared and the assay was performed using the LightShift Chemiluminescent EMSA Kit (Thermo#20148), as per the manufacturer's instructions. Biotin–EBNA Control DNA, Unlabeled EBNA DNA, and Epstein-Barr nuclear antigen (EBNA) extract were used as controls.

The biotin-labeled *crq* promoter DNA containing the binding site was annealed by PCR to form oligos using the following primers: *crq* forward GGACGCGTACGGTTTTGATAGT TCCGATTGTG, and reverse CACAATCGGAACTATCAAAACCGTACGCGTCC. The biotin–unlabeled *crq* promoter DNA containing the binding site was annealed by PCR to form oligos using the following primers: *crq* forward CGTACGGTTTTGATAGTTCCG and reverse CGGAACTATCAAAACCGTACG.

Using 2 μL nuclear extract per 20 μL binding reaction, nuclear proteins from the HA-Srp-transfected S2 cells were extracted using the NE-PER Nuclear and Cytoplasmic Extraction Reagents (Thermo #78833). Anti-HA (rabbit, CST, 1 μL per reaction) was used to observe the supershift of anti-HA/HA-Srp/DNA. Srp-GST and Bfc-His were expressed and purified as previously described [68]. The Bfc pET28a construct was constructed by PCR amplification using the following primers: Bfc-his forward tggacagcaaatgggtcgcATGAGAATGGACAAGT CGGA, Bfc-his reverse tggtggtggtggtggtgTGTCTTCTTGGCTGGCACG. The Srp pGEX KG construct was constructed by PCR amplification using the following primers: Srp-GST forward ggtggtggtggaattcATGACGAAAACGACAAAGCC and Srp-GST reverse tcacgatgaat taagcttCTCCATCTTCACCACTG.

## Statistical analysis

Statistical *P*-values were determined using the one- or two-way analysis of the variance or the Student's *t*-test. "*" indicates $P < 0.05$; "**" indicates $P < 0.01$; "***" indicates $P < 0.001$, "****" indicates $P < 0.0001$. The error bars represent the SEM.

## Supporting information

**S1 Fig. ACs were successfully generated from S2 cells, and the RNAi knockdown efficiency was confirmed by qPCR. A.** S2 cells and ACs were stained with annexin V-FITC and propidium iodide (PI). Scale bar = 20 μm. ACs were obtained from S2 cells after Actinomycin D induction for 18 h, as described in the Materials and Methods section. **B.** The apoptosis rates of S2 cells and ACs were analyzed by flow cytometry. **C.** Western blotting detected the Dcp1 protein levels in S2 cells and ACs. Actin was used as the loading control. **D.** The relative

mRNA levels of 24 genes in RNAi-treated S2 cell lines were quantified by qPCR and compared to those of control-treated S2 cells; *RpL32* was used as the internal control. Statistical significance was assessed using the two-way ANOVA.
(TIF)

**S2 Fig. *CG9129* and *crq* regulate each other at the transcriptional level during the clearance of ACs in S2 cells. A.** The *CG9129* mRNA levels were quantified by qPCR in Crq-overexpressing S2 cells (Crq-GFP) and control S2 cells. The *crq* mRNA levels were also quantified by qPCR in CG9129 over-expressing S2 cells (CG9129-flag) and control S2 cells. 0 h refers to S2 cells in the absence of ACs; 6 h refers to S2 cells co-incubated with ACs for 6 h. *RpL32* was used as the internal control. **B.** The *CG9129* mRNA levels were also determined in RNAi-treated S2 cells (control and for the knockdown of *crq*) after co-incubation with ACs for 6 h, by qPCR. Statistical significance was assessed using the two-way ANOVA (**A**) or Student's *t*-test (**B**).
(TIF)

**S3 Fig. Bfc regulates the expression of Crq *in vivo*. A.** The relative *crq* mRNA levels in stage 7–14 wild type embryos were quantified by qPCR; significant differences were observed compared to the expression in stage 7 embryos. **B.** The *bfc* mRNA levels in stage 7–14 wild-type embryos were quantified by qPCR; significant differences were observed compared to the expression in stage 7 embryos. **C.** The relative *crq* mRNA levels were quantified and compared in the same stage of wild type and *bfc*$^{ko}$ mutant embryos by qPCR. **D.** Western blot analysis was used for the detection of the Crq protein levels in stage 11–14 embryos of wild type and *bfc* mutant flies. Actin was used as the loading control. **E.** The relative Crq protein levels in stage 11–14 wild type, and *bfc*$^{ko}$ mutant embryos were analyzed via western blotting. The protein levels were normalized to those of actin. Significant differences were observed in comparison with stage 11 wild-type embryos; n = 3. **F.** The relative *crq* and *bfc* mRNA levels were quantified by qPCR in stage to 10–13 wild-type *w*$^{1118}$ and *H99* (absence of apoptosis) embryos. **G.** The *crq*, *bfc*, and *drpr* mRNA levels were quantified by qPCR and normalized to the *Rpl32* mRNA levels. "*"/"#"/"※" represent the significant differences in the context of *crq/bfc/drpr* expression, respectively. "##" indicates P<0.01; "###" indicates P<0.001; "####" indicates P<0.0001; "※※" indicates P<0.01. **H.** The *drpr* mRNA levels were quantified by qPCR in wild-type S2 cells and *bfc*-RNAi S2 cells in the presence or absence of ACs. Statistical significance was assessed using the one-way ANOVA(**A, B, E, F, H**) or two-way ANOVA (**C, G**).
(TIF)

**S4 Fig. *bfc* knockout reduced *crq* expression without affecting macrophage development in the embryos. A.** The *crq*, *drpr*, and *simu* mRNA levels were quantified by qPCR in stage 13 wild-type, *bfc*$^{ko}$, and *srp* mutant embryos. **B.** Macrophages were stained with anti-CRQ (green) antibodies, and ACs were stained with anti-Dcp1 (magenta) antibodies in stage 13 wild-type and *bfc*$^{ko}$ embryos. Scale bar = 20 μm. **C.** Graph showing the number of macrophages ± SEM for genotype of wild type, *bfc*$^{ko}$, *srp-Gal4>UAS-GFP* and *bfc*$^{ko}$; *srp-Gal4>UAS-GFP*. Statistical significance was assessed using the two-way ANOVA (**A**) or one-way ANOVA (**C**).
(TIF)

**S5 Fig. Bfc is required for efferocytosis of embryonic macrophages. A.** Z-stack of eight images from a single macrophage of wild type and *bfc*$^{ko}$ embryos taken at intervals of 1 μm. Scale bar = 5 μm. **B-B'.** Control *srp-Gal4>UAS-GFP* embryo macrophages were labeled with *srp-Gal4>UAS-GFP* (green); anti-Dcp1 was labeled in magenta. **C-C'.** *bfc*$^{ko}$; *srp-Gal4>UAS-GFP* embryo macrophages were labeled as *srp-Gal4>UAS-GFP* (green); anti-Dcp1 was labeled in magenta. Scale bar = 20 μm. **D.** Graph showing the mean PIs ± SEM for each genotype

**(A-C).** Statistical significance was assessed using the one-way ANOVA.
(TIF)

**S6 Fig. *bfc^{MI02020}* mutants show a defective efferocytosis of ACs. A.** The *crq* and *bfc* mRNA levels were quantified by qPCR using wild-type and *bfc^{MI02020}* embryos. **B.** Graph showing the mean PI ± SEM for each genotype in (**C**). **C.** Anti-CRQ (green) and anti-Dcp1 (magenta) antibodies were used to stain the apoptotic bodies in macrophages of stage 13 wild-type, *bfc^{MI02020}*, and *bfc^{MI02020}* re-expressing *UAS-bfc* (under the control of a *crq-Gal4* driver) embryos. The white arrowheads point to non-engulfed ACs. Scale bar = 20 μm. **D.** Anti-CRQ (green) and anti-Dcp1 (magenta) antibodies were used to stain the apoptotic bodies in macrophages of stage 13 wild-type, *bfc^{ko}*, and *bfc^{ko}* re-expressing *UAS-crq* (under the control of a *crq-Gal4* driver) embryos. The white arrowheads point to non-engulfed ACs. Scale bar = 20 μm. **E.** Graph showing the mean PI ± SEM for each genotype in (**D**). **F.** Anti-CRQ (green) and anti-Dcp1 (magenta) antibodies were used to stain the apoptotic bodies in heterozygous macrophages of stage 13 *bfc^{ko}*/+ and *srp*/+ embryos. The white triangles point to non-engulfed ACs. Scale bar = 20 μm. Statistical significance was assessed using the two-way ANOVA **(A)** or one-way ANOVA **(B, E)**.
(TIF)

**S7 Fig. *bfc* and the *crq* promoter interact with the ZnF GATA domain of Srp. A.** Schematic diagram showing the domains of Srp and the truncations generated to examine the potential Bfc-Srp interactions. **B** and **C.** The interaction between Bfc and Srp was examined using the yeast two-hybrid (**B**) and Co-IP assays (**C**). **D.** Schematic diagram showing the C4^{M} mutants with the substitution of eight cysteines by arginines in the C4 motif of Srp. **E.** The yeast two-hybrid assay showing the interaction between Bfc and the C4^{M} Srp mutant. **F.** The yeast one-hybrid assay shows the lack of interaction between the *crq* promoter C4^{M} and the C4^{M} Srp mutant.
(TIF)

**S8 Fig. Mutations in the GATA site result in the weaker activation of Crq expression in *Drosophila* embryos. A.** Anti-Flag (green) and anti-Dcp1 (magenta) antibodies were used to stain macrophages and ACs of stage 13 wild-type, *bfc* mutant re-expressing UAS-CrqFlag (under the control of a *crq-Gal4* driver), and wild-type re-expressing UAS-CrqFlag (driven by mutated *crq-Gal4*) embryos. Scale bar = 20 μm. **B** and **C.** The Flag protein levels were determined via western blotting for each genotype in **(A)**. Actin was used as the loading control. **D.** Quantification of the Flag protein levels in **(B)** and **(C)** after normalization to those of actin (n = 3). **E.** The Crq protein levels were detected via western blotting in the context of *w^{1118}*, *bfc^{ko}*, *bfc^{ko}* mutant re-expressing UAS-CrqFlag (driven by *crq-Gal4*), and wild-type re-expressing UAS-CrqFlag (driven by *crq-Gal4*) samples. Actin was used as the loading control. **F.** Quantification of the Crq protein levels in **(E)** after normalization to those of actin (n = 3). Statistical significance was assessed using the Student's *t*-test.
(TIF)

**S9 Fig. Bfc activates *crq* promoter to drive GFP protein through the GATA site in *Drosophila* embryos, while Bfc has no influence on Srp level. A-A".** Anti-GFP (green) and anti-Dcp1 (Red) antibodies were used to stain macrophages and ACs of stage 13 embryos in *crq-Gal4>UAS-GFP*(**A**), *crq^{M}-Gal4>UAS-GFP*(**A'**), and *crq^{M}-Gal4>UAS-GFP; bfc^{ko}*(**A"**). Scale bar = 20 μm. **B.** The GFP protein levels were determined via western blotting using *crq-Gal4>UAS-GFP*, *crq^{M}-Gal4>UAS-GFP*, and *crq^{M}-Gal4>UAS-GFP; bfc^{ko}* embryos. Actin was used as the loading control. **C.** Quantification of the protein levels in **(B)** after normalization to actin; n = 3. Statistical significance was assessed using the Student's *t*-test. **D.** The Crq and

Srp protein levels were determined via western blot analysis in blank, Bfc-overexpressing, and *bfc-RNAi*-treated S2 cells in the presence or absence of ACs. Actin was used as the loading control. **E.** Quantification of the protein levels in (**D**) after normalization to actin; n = 3. Statistical significance was assessed using the one-way ANOVA(**C**) and two-way ANOVA(**E**).
(TIF)

**S10 Fig. The yeast one-hybrid assay revealed that Srp and Stat92e bind to the *crq* promoter and *drpr* promoter, respectively. A.** Schematic diagram showing the domain of the *crq* promoter used for the yeast one-hybrid assay. **B**. The yeast one-hybrid assay shows the interactions between Srp-AD, Bfc-AD, Stat92e-AD, and the *crq* promoter. **C.** Schematic diagram showing the domain of the *drpr* promoter used for the yeast one-hybrid assay. **D.** The yeast one-hybrid shows the interactions between Srp-AD, Bfc-AD, Stat92e-AD, and the *drpr* promoter.
(TIF)

**S1 Data. Differentially expressed genes from comparisons of 6h versus 0h, 12h versus 0h, and 6h versus 12 h, respectively.**
(XLSX)

**S1 Table. RNA-seq data of the 50 filtered genes.** Representative genes were selected across a range of expression levels.
(DOCX)

**S2 Table. qPCR primers used to obtain the data represented in Fig 2.**
(DOCX)

**S3 Table. dsRNA primers used to obtain the data represented Fig 2.**
(DOCX)

**S4 Table. Recombinant DNA constructs used in this study.**
(DOCX)

**S5 Table. Raw data for the statistical graphs used in all figures.**
(XLSX)

## Acknowledgments

We thank the Bloomington Drosophila Stock Center for the *Drosophila melanogaster* strains. We would like to thank Editage (www.editage.cn) for English language editing.

## Author Contributions

**Conceptualization:** Qian Zheng, Hui Xiao.

**Data curation:** Qian Zheng, Ning Gao, Qiling Sun, Hui Xiao.

**Formal analysis:** Qian Zheng, Ning Gao, Hui Xiao.

**Funding acquisition:** Hui Xiao.

**Investigation:** Qian Zheng, Ning Gao, Qiling Sun, Xiaowen Li, Yanzhe Wang, Hui Xiao.

**Methodology:** Hui Xiao.

**Project administration:** Hui Xiao.

**Resources:** Hui Xiao.

**Software:** Hui Xiao.

**Supervision:** Hui Xiao.

**Validation:** Qiling Sun, Hui Xiao.

**Visualization:** Hui Xiao.

**Writing – original draft:** Qian Zheng, Ning Gao, Qiling Sun, Hui Xiao.

**Writing – review & editing:** Qian Zheng, Ning Gao, Qiling Sun, Xiaowen Li, Yanzhe Wang, Hui Xiao.

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
