## [Decision Letter · Decision Letter 0]

17 Mar 2021

Dear Dr Xiao,

Thank you very much for submitting your Research Article entitled 'Bfc, a novel Serpent co-factor for the expression of Croquemort, regulates efferocytosis in Drosophila melanogaster' to PLOS Genetics.

Your manuscript has now been reviewed by three expert referees. These reviewers expressed major reservations and felt that more work was required to substantiate the conclusions drawn. In addition, some of the results need better explanation and re-evaluation of data, and the methods section and the figure legends must be re-written. English usage and grammar should be improved throughout the manuscript including the appropriate use of abbreviations.

We would like to re-consider a substantially revised version of the manuscript if the reviewers' concerns are adequately addressed. We cannot, of course, promise publication at that time.

If you decide to revise the manuscript for further consideration at PLOS Genetics, please aim to resubmit within the next 60 days, unless it will take extra time to address the concerns of the reviewers, in which case we would appreciate an expected resubmission date by email to plosgenetics@plos.org.

[LINK]

We are sorry that we cannot be more positive about your manuscript at this stage. Please do not hesitate to contact us if you have any concerns or questions.

Yours sincerely,

Estee Kurant

Guest Editor

PLOS Genetics

Gregory P. Copenhaver

Editor-in-Chief

PLOS Genetics

Reviewer's Responses to Questions

**Comments to the Authors:**

Reviewer #1: In their manuscript, Zheng et al. identify CG9129/Bfc as a gene required for proper expression of the apoptotic receptor Croquemort in Drosophila S2 cells and embryonic macrophages. Consistent with this observation, they found that Bfc loss is associated with a decrease in apoptotic cell phagocytosis. They further show that Bcf interacts with the GATA transcription factor Srp and their results suggest that Bfc could promote the direct binding of Srp to crq, thereby favoring crq transcription.

These findings are of potential interest but several conclusions are poorly substantiated. Importantly too, key controls and information are missing. Moreover, the level of analysis is rather limited and, in my opinion, beyond the expectations for PLoS Genetics.

Finally, the manuscript is poorly presented: several pieces of results are not well described or presented in a logical order, the figure calling is not accurate and the figure layout is often difficult to grasp, the text is plagued with typos and grammatical errors, and the “methods and materials” section is not appropriately written.

The comments below may help the authors to improve their manuscript.

Results sections:

- The authors should mention how many replicates were used for the RNAseq. Page 4: it does not make any sense to write that ±9000 genes (i.e. ±70% of the genes encoded by the genome) are differentially expressed at the different time points; differential expression must be defined with clear cutoff (fold change and p values, as used page 5).

- Page 5: “engulfing S2 cells displayed changes in multiple transcriptional programs”: where are the analyses /tables for such conclusion?

- Page 5 / Figure 2: the criteria used to select the 24 genes are not very clear; some genes that were kept/eliminated do not seem to strictly follow the threshold/statistical significance criteria (eg: PGRP-SA, CG12112…).

- How the author explain that several genes tested by RT-qPCR did not behave as in the RNAseq experiment ?

- Page 5 / Fig. 2c: it is not clear how efferocytosis efficiency is measured. Also the effect of crq knock-down seems very mild and it is thus questionable whether this assay is a good readout.

- Fig 2d: at which time point is crq expression assessed?

- Page 6/7: the parallel analyses for the two alleles of Bfc should be presented at the same time.

- Why is there more crq mRNA in stage 11 bfc null embryos (fig S3e)? This is not consistent with the author hypothesis.

- The legend is missing for Fig S3F.

- Figure 3a: the effect of Bcf RNAi on S2 cells AC engulfment is not quantified.

- Figure 3e: why is acridine orange staining so low in Bcf KO embryos?

- Page 7: the authors can not conclude that Bfc KO did not affect macrophage development. They should use other markers than Crq to quantify macrophage number / repartition. Actually, with Crq immunostaining (Fig 3f), it seems that there are less macrophages in the head region (and the level of Crq in each cell does not seem reduced as compared to control!).

- Page 8: “down-regulation of crq in the absence of Bfc caused the defect in AC clearance”: this is an important point that needs to be formally addressed. In addition to their data that support this hypothesis, the author should rescue Bfc KO by re-expressing Crq and assess AC clearance.

- Fig 4a: the author should show a higher magnification of Bfc immunostaining together with DAPI or another nuclear marker to demonstrate that Bfc is in the nucleus in vivo. The quality of the present image is clearly insufficient.

- Fig 4c: an important negative control is missing (IP with anti-Flag on S2 cells transfected only with HA-Srp). The authors should also try to perform the IP with anti-HA and with the endogenous Bfd (using the anti-Bfd).

- Fig S5d: this may not be “endogenous” Bfc in S2 cells since these cells are transfected with Bfc-FLAG.

- the immunostainings with Bcf and Flag antibodies in Fig S5d look very different from the one shown Fig 4d (GFP-Bfc). This casts doubt on the results. The immunostainings in 4d is very strange as Srp is not known to form such kind of foci. Again some negative controls are missing (single transfections + double immunostaining).

- Page 9 / Figure 5: the authors claim that single heterozygous did not show differences but these results are not shown/quantified. They must be shown.

- Page 9: the last paragraph is unclear and not connected with Srp/Bfc interaction.

- Page 10: the two-hybidrid experiments described here are not displayed Fig S5 but Fig S6. Moreover, there is a clear decrease in the binding of Srp to Bfc when the Cys are mutated to Arg.

- Page 10 / Figure 6: the authors claim that they assess Bfc and STAT92E (why?) binding to crq regulatory sequence but the data are not shown.

- Page 10/ last line: it is inaccurate to state that “crqM-Gal4 failed to activate transcription”. This driver is just weaker than crq-Gal4. In addition, the authors should test whether crqM-Gal4 activity is (still) affected by Bfc KO and/or srp KO.

- The ChIP experiments results are weird. Anti-HA ChIP with non-transfected S2 cells would be a better control. Also it is very unlikely that you can get around 40% of Srp ChIPed to crq as compared to input (even with Histones, people usually end up with a few % of efficiency).

- The EMSA experiments are ill-described and not performed properly (100% of the probe is shifted in the presence of GST-Srp). The evidence that Bcf promotes Srp binding are very weak. Bfc dose-dependant effect should be demonstrated in EMSA; and reciprocal experiments with Bfc knock-down should be performed. Also it should be checked by western blot that Srp level is not affected upon Bfc overexpression

TheM&M does not give sufficient details concerning many experiments. Notably:

- none of the plasmids used in the transfection assays are described.

- The RNAseq analysis is poorly described.

- The efferocytosis assays and quantification is not clearly presented.

- Details concerning the anti-Bfc antibody production are missing.

- The EMSA assays are not presented with sufficient details (eg: quantity of poly dIdC, fold excess for crq probe competition…).

- Protocols for nuclear extract preparation and coIP protocol are not given.

Other comments:

- The RNAseq data should have been deposited in a public database and a link provided for reviewer access if the data have not been released yet.

- the standard deviations for the control situations are apparently missing in several graphs (for example: Fig 1a, c, 1e, 2a, 2b, 2d, 2e, 5f, 6c…); it is thus impossible to evaluate the significance of the differences between control and test situations.

- It is generally accepted that figure calling follow their order of appearance in the main text, which is not always the case here (e.g.: Fig 3a, Fig 4c /d....).

- Some figures / part of figures are not presented in the text (for example figure 6d right panel).

- The arrangement of the panels could be improved to follow a more logical order and facilitate the interpretation by the reader. For example, Figure 2: the “positive” and “negative” genes could be displayed in 2a and 2b respectively and the order of the genes should be kept constant (alphabetical?) in the different panels. Figure 5: place the protein level quantifications below or beside the corresponding western blots and the phagotytic index graphs next to the corresponding images).

- It is customary to present lateral views of embryos dorsal side down.

- There is a number of recurrent typos; notably: space missing between word and bracket when citing a paper, um instead of µm, CHIP instead of ChIP.

- The material and methods sections should be rewritten more clearly, not as a lab protocol, and taking care of correcting grammatical errors as well as scientific mistakes (example: DNA primers do not have a N-terminal but a 5’)

Reviewer #2: Zheng et al describe a new regulator of efferocytosis in Drosophila, bfc. Bfc physically and genetically interacts with the GATA factor Serpent, a key transcription factor used in specification of Drosophila blood cells. Bfc was identified via transcriptional profiling of S2 cells treated with dying cells in vitro. A number of other interesting genes that are modulated via exposure to apoptotic cells are described but not followed up further and these too will be interesting to the field.

A subset of these genes regulate levels of croquemort, a gene initially identified on the basis of a role in apoptotic cell clearance. From these genes Bfc is characterised further and shown to regulate crq at both the mRNA and protein level. Bfc interacts with Serpent in vitro and within cells and there are convincing genetic experiments showing that it plays a role in Crq levels in vivo. The authors suggest Bfc acts as a co-factor of Serpent to promote Crq expression in response to macrophages facing apoptotic cells. The data suggest that Serpent regulates crq expression via a GATA site in the crq promoter. Less clear is the exact mechanism via which Bfc exerts its effects and I think that part of the manuscript needs to be improved.

The paper is very interesting and contains significant amounts of work. However, further work is required to strengthen and support the conclusions that have been made, although many of my comments can be addressed with careful re-writing or re-analysis. The manuscript also needs substantial work to aid the reader in following the manuscript and to ensure reproducibility. Please see my detailed comments below. I hope the authors will not feel disheartened by the number of comments I have made as I feel generally they should be fairly easy to address and the majority do not require further experiments to be carried out.

Major points

--------------

1. I am afraid that I found the manuscript hard to follow in many places. The figure legends and methods are insufficiently detailed and in many places the results are not described well enough to follow what has been done. The methods are written in a note form that is closer to a protocol than the methods section of a publication. Therefore, the entire manuscript would benefit from careful rewriting to ensure sufficient detail is included to aid the reader. This is particularly the case in the section relating to Figure 6 and 7. Additionally, in some places the language used is not grammatically correct or could be improved to help understanding.

2. The statistics section in the methods is too brief and it appears that inappropriate statistical analyses have been used in many of the figures: according to the methods section it seems that multiple t-tests have been used to compare >2 experimental conditions. Details of the statistical tests used are not described in the figure legends. Therefore, all numerical data and the statistical tests used to compare experimental groups need careful checking and (if needed) a different statistical analysis performing. This may change whether certain data remains statistically significant, so conclusions may need to be modified.

3. While I am largely convinced that Srp and Bfc interact and that Srp is binding the crq promoter via a GATA site, I am not convinced that the mechanism by which Bfc is working with Srp is correct or supported by the data presented. This model and Figure 7 are not explained very clearly in the manuscript. I am not sure the data supports the idea that Bfc increases the amount of Srp bound to the crq promoter. However, the way in which the IP and EMSA data are described does not help the reader. Can the authors show that the presence or absence of Bfc regulate the interaction of Srp with the crq promoter (e.g. via ChIP)? For instance: P11 “Bfc interacts with the zinc-finger domain of transcription factor Srp as a co-factor to enhance Srp binding to the crq promoter to elevate its expression and induce efferocytosis in Drosophila melanogaster.” I am not sure that the EMSA data is consistent with this conclusion, but it is not adequately described in the manuscript.

Aditionally, it is not clear what the controls on the left of Fig 6d are or what the panel on the furthest right is demonstrating, as this is not explained anywhere. Authors claim an increase in the migration efficiency of the Srp-DNA complex in the presence of Bfc. Is this the very faint band that can just about be made out below the main band? If so, it is not clear what this means in terms of the mechanism and does not appear to be consistent with Bfc binding to the DNA in concert with Srp, as that should decrease mobility of the band. It is thus not clear how the interaction of Bfc with Srp is functioning to increase Crq levels from this data. Also, why is the Bfc + Srp condition repeated so many times on this gel?

On page 14: “Furthermore, Bfc-Srp interaction occurred through the ZnF GATA domain and that the presence of Bfc increased the ability of Srp to bind the crq promoter.” I don’t think this is substantiated by the data – if it refers to the slight increase in intensity in Fig 6d (not quantified) and this seems an inappropriate conclusion as there is not an excess of DNA (no free probe visible at the bottom). Therefore, probe is rate limiting for complex formation and so adding a fraction more in one lane would change intensity.

4. Authors claim to have analysed STAT92E, but no data is presented: “To detect the physical interactions between regulatory transcription factor proteins and crq in the genome, we then performed yeast one-hybrid (Y1H) assays(Reece-Hoyes and Marian Walhout, 2012) between Bfc, Srp, Stat92E, and the crq promoter locus, respectively.”

5. Apoptotic cell clearance is only analysed in the head. Can defects in apoptotic cell clearance be detected elsewhere in the embryo in bfc mutants? Dcp-1/Crq staining is only shown for the head region. Acridine orange staining looks like it may be less affected in the absence of Bfc along the ventral regions of the embryo. This is particularly important due to the emerging idea that different subtypes of macrophage may be present in the fly (e.g., Cattenoz et al., 2020; Tattikota et al., 2020).

6. Is anti-Crq the best marker to stain and visualise macrophages accurately in bfc mutants if Crq protein levels are so decreased in the absence of Bfc (Figure S5c and other similar figures)? I have always found it difficult to see how phagocytic index is accurately measured using anti-Crq staining in comparison to, for example, use of srpHemo-GAL4,UAS-GFP and other such tools; it is hard to see where one cell begins / ends and how many cells are present in these images. It is also hard to see what is going on due to the overlap of red and green staining in these images (can single channel images be displayed with the merge?) Also, please consider using green/magenta rather than green/red due to those suffering from colourblindness, especially if separate channels are not to be displayed. What do the white dotted circles represent? Individual cells? Lastly, the use of anti-Crq makes it difficult to score these images blindly as mutant and wt/rescues can easily be discriminated and this appears to be quite subjective scoring anyway (precise details are missing from the methods). N.b., these comments apply to all figures that show anti-Crq/anti-Dcp-1 staining.

7. Figure 4d shows Bfc is nuclear in S2 cells. Can a nuclear marker/dye and higher magnification imaging show a similar localisation in macrophages in the embryo? What happens to Srp localisation on RNAi of Bfc from S2 cells? This should be fairly easy to do with the reagents already in hand and may give an insight into how the Bfc-Srp interaction works to regulate Crq levels (especially as I am not sure about the EMSA data).

8. Why are immune genes like cecropin coming up in the transcriptional profiling? This seems counterintuitive to the silent clearance of apoptotic cells, as it suggests immune activation. We ourselves have had issues generating apopototic cell samples via similar methods – we obtain apoptotic cells but also a large amount of necrotic cells. The AC samples used/method therefore require characterisation/validation (e.g., caspase activity probes, annexin, PI – all widely available). It therefore is possible that some of the genes that are being upregulated are in response to necrotic cells present in the samples in addition to the apoptotic ones. The AC samples should be characterised and if necrotic cells are present the conclusions need to be modified to acknowledge that the transcriptional changes observed may not be specific to contact with apoptotic cells.

9. In general, the discussion needs to discuss the findings in a broader context, as it mostly simply repeats the conclusions/findings of this paper. In particular there is barely any discussion of the work in relation to other organisms (only 1 mammalian paper is cited). For instance, do homologs of Bfc exist in vertebrates? Is this a fly specific mechanism? A careful re-write of this section will make it much more appealing and relevant for non-Drosophilists.

Minor comments

------------------

1. Figure 1d-e what does % phagocytosis mean? I presume % of S2 cells phagocytosing AC – please make clearer in legend and methods.

2. I am not sure that it is possible to make the conclusion that “as an engulfment receptor, Crq contributed to early recognition of ACs, while engulfment at later stages likely depended on other regulatory factors to mediate further AC clearance.” It is not clear whether the internalised AC were internalised at earlier times and have not yet been degraded or have been internalised more recently. In my opinion to test this properly you would need to stimulate with AC, remove excess AC and then challenge again, or something similar (if this was actually how the experiment was carried out it is not clear from the results, methods or figure legend). I don't propose this experiment is repeated as suggested, as this doesn't seem a key point, just that the conclusions are modified.

3. What are the experimental conditions for Figure 2c,d? This is not clear from the text, methods or legend. I presume this is with AC treatment, but for how long etc?

4. The effect of knockdown of CG9129/bfc is far stronger than for crq itself (Figure 2c), what other processes may be affected given bfc seems to play no role for SIMU and drpr? It may be nice to speculate on this in the discussion.

5. Figure 3a appears after Figure 3c-d. Authors may want to reorder the text / figure slightly to improve how the manuscript reads. There are quite a few panels not cited (e.g. Fig S3e) in the text or mistakes in which figures are being cited throughout the manuscript.

6. Not all of the inset can be seen in the main field of view for the central panel of Figure 3a, while the zooms could have been a bit bigger. I also struggled to see where the regions were taken from (in fact I am unable to see where the zoom for bfc RNAi came from at all for that field of view).

7. Please provide details of how the mouse anti-Bfc was generated (what epitope was used etc). Authors should check for other methods that are not described in the paper.

8. Figure 4a - why is the green channel so bright in the bfc[ko] embryo? Have these embryos been imaged using the same settings/enhanced similarly? Why is the green channel so faint in the boxed region of the bfc[ko] panel compared to the main image? Individual channels should be shown and advisable to use magenta in place of red for merges. Also, it would be better not to inset zoomed regions over the posterior of the embryo as this prevents the reader from analysing expression in posterior macrophages.

9. References required for this statement on p8: “Previous researches have demonstrated that several signal pathways regulated the expression of engulfment receptor Drpr, such as d-JNK, PI3K and insulin.”

10. page 10 – some confusion / mix ups for figure labelling for Figures S5-7. Not all panels are cited in the text.

11. Why does bfc not work to regulate the amounts of other Srp-dependent receptor genes? Authors suggest other co-factors regulate Srp’s activity at the drpr and simu promoters but I don’t think their model is sufficient to explain why Bfc would not assist at those promoters given their regulation by Srp. Perhaps it would be clearer with more explanation?

12. P13 “These results imply that the expression of phagocytotic receptors could be stimulated by the presence of excessive ACs to improve the phagocytotic activity of macrophages.” This has been shown by a number of other labs in the fly (e.g., Wood and McCall) in flies and is well-documented in vertebrates. These and other papers should be cited and this new mechanism contrasted with how this is regulated in other systems.

13. Authors may wish to include Han et al. (2014) and Guillou et al. (2016) in the discussion since these suggest functions for Crq other than a role as a receptor for apoptotic cells.

14. Supplemental tables 3 and 4 are not cited in the manuscript.

Reviewer #3: The review is uploaded as an attachment.

**Have all data underlying the figures and results presented in the manuscript been provided?**

Reviewer #1: **No: **No GEO accession number/link to a public database was provided for the RNAseq data. None of the values used to build the graphs were provided as supporting information.

Reviewer #2: **No: **I am not sure that the RNAseq data has been or will be uploaded to a public repository. It should be uploaded on acceptance with the location contained within the paper in the final published form. Numerical data not supplied in current submission.

Reviewer #3: Yes

PLOS authors have the option to publish the peer review history of their article (what does this mean?). If published, this will include your full peer review and any attached files.

Reviewer #1: No

Reviewer #2: No

Reviewer #3: No

---

## [Decision Letter · Decision Letter 1]

7 Jul 2021

Dear Dr Xiao,

Thank you very much for submitting your revised version of the Research Article entitled 'Bfc, a novel Serpent co-factor for the expression of Croquemort, regulates efferocytosis in Drosophila melanogaster' to PLOS Genetics.

The manuscript was fully evaluated again at the editorial level and by three independent peer reviewers.  Two of them now ask minor revision whereas Reviewer #1 requests additional experimentation which the editors agree is necessary to support the central conclusions of the study. In addition, English usage and grammar still need improvement throughout the manuscript - we find that authors often benefit from having their manuscripts proof read by a native English speaking colleague, or by using a reputable copy editing service.  

If you decide to revise the manuscript for further consideration at PLOS Genetics, please aim to resubmit within the next 60 days, unless it will take extra time to address the concerns of the reviewers, in which case we would appreciate an expected resubmission date by email to plosgenetics@plos.org.

[LINK]

We are sorry that we cannot be more positive about your manuscript at this stage. Please do not hesitate to contact us if you have any concerns or questions.

Yours sincerely,

Estee Kurant

Guest Editor

PLOS Genetics

Gregory P. Copenhaver

Editor-in-Chief

PLOS Genetics

Reviewer's Responses to Questions

**Comments to the Authors:**

Reviewer #1: In their revised manuscript, Zheng et al. have answered some of the concerns raised by the 3 referees and have improved the overall quality of their work. Yet, beside the presentation of the data and M&M that needs to be improved, several conclusions are not well substantiated and some experiments are still not performed adequately. In addition, some of the modifications introduced during the revision do not speak in favor of a rigorous analysis of the results by the authors.

Thus, even though the authors made important modifications to their manuscript, I still think that it does not reach the standards of quality expected for publications in PLoS Genetics.

Some specific comments to the authors.

Major points:

- The statistical significance of many results shown in Figure 2 and S1 have changed as compared to the original version of the manuscript even though they are based on the same numerical data. While I understand that the authors have now used different methods (Annova), I’m very skeptical as some points “gained” one or even two stars of significance and other lost up to three stars!! (e.g: pgrpSA at 12h was “ns” and is now p<0.01; CG8907 at 6 and 12h were p<0.01 and are now ns!). This casts important doubts on the analyses of the data by the authors.

- The authors claim that bfc mutation does not affect macrophage development but the data are still not convincing (S4B-C). Notably, the authors used srp-GAL4>GFP to label macrophages and they observed around 100 macrophage per embryo in wildtype and bfc mutant contexts. This is far less than expected as wildtype embryos contains several hundreds of macrophages!

- The rescue experiment with crq-GAL4 UAS-Crq was performed in the hypomorphic bfc mutant and not in the null context (S6). As the authors showed that Crq induces bfc transcription, it is thus difficult to conclude that it the restauration of Crq level rather than the restauration of Bfc level which is responsible for the rescue.

- The impact of apoptosis/ACs on Srp/Bcf-mediated regulation of crq is not clearly assessed. Notably, the authors do not formally test that apoptosis is important for the induction of Bfc and/or Crq expression in vivo. Actually, line 213/FigS3B, the authors show that bcf expression is upregulated as early as stage 9, i.e. before the first wave of apoptosis (stage 11). The conclusion lines 235-236 is not substantiated by the in vivo data. The authors could use the H99 deletion to prevent apoptosis in the embryo and assess whether this affects bfc and crq induction. Thus far, the conclusion line 505-506 is not really sustained.

- The hypothesis that Bfc promotes Srp binding to crq regulatory region is still poorly substantiated. First, the results of the EMSA are not convincing: the supershift with Bcf/Srp is barely detectable and does not really increases with Bcf dose. The binding of Srp to crq probe seems highly variable (cf 2E and 2E’); it may increase with Bcf but this could reflect a mere stabilization due to increased protein concentration in the reaction (increased Srp binding is also observed when anti-HA antibody is added to the reaction in 2E’). Also, line 7 does not seem to contain any probe, and the competition with the unlabeled crq probe does not seem to work in panel 2E. Second, the authors did not test the influence of Bfc on Srp recruitment to crq region by ChIP and they did not assess whether Bfc over-expression/knock-down affects Srp protein level (by western blot). Again these experiments (ChIP, luciferase assays, western blots) should be performed in S2 cells exposed to ACs to be able to conclude on the molecular mechanisms by which ACs control crq expression (lines 469-470).

Minor points:

- Lines 120-123: I don’t see a particularly logical link between ACs clearance and bacteria uptake. Even if there is one, the level of Crq is lower at 12h than at 6h but efferocytosis is higher at 12h

- In the discussion: lines 550-552 and 562-565 are identical.

- Lines 560: references are missing concerning the regulation of crq by Srp. Also, when discussing the role of Ush/Srp (lines 538-541): the induction of crq by Srp was shown to be repressed by Ush in Waltzer et al EMBO J 2002.

- Some parts of the material and methods are still written in a note form closer to a protocol than the methods sections of a publications (e.g.: lines 697-699; 723-728; 746-751; 758-770; 778-779

- I did not find the GO annotations data for the RNA-seq in the supplemental data (lines 649).

- It is still not clearly mentioned in the M&M that the authors generated not only a crqM-GAL4 but also the crq-GAL4 “wildtype” version, integrated at the same site an in the description of the results/figure legends, it should be clearly mentioned when the authors use Bloomington crq-GAL4 or their crq/crqM-GAL4 lines.

- In their answer to reviewer 1, the authors claim that they could not recombine crqM-GAL4 with bfc or srp; yet according to their M&M crqM-GAL4 is on chromosome II, so it shouldn’t be a problem. And if it is on chr III, it should be possible to obtain recombinant at least with one of the two genes (bfc or srp) as they are on opposite chromosomal arms.

- Figure 4D: the negative controls (right panel) should have been performed in the same experiment and not added afterward to the original coIP (left panel).

- Figure S7C: part of the anti-HA western blot (lanes HA-Srp4) is visibly spliced from another gel.

- Many typos need to be corrected (e.g.: line 19: Srp shouldn’t be in italic, line 34: (ACs) => ACs; line 41: induce => induces…).

Reviewer #2: Uploaded as an attachment.

Reviewer #3: The review is uploaded as an attachment.

**Have all data underlying the figures and results presented in the manuscript been provided?**

Reviewer #1: **No: **Numerical data that underlies graphs are not provided.

Reviewer #2: Yes

Reviewer #3: Yes

PLOS authors have the option to publish the peer review history of their article (what does this mean?). If published, this will include your full peer review and any attached files.

Reviewer #1: No

Reviewer #2: No

Reviewer #3: No

---

## [Decision Letter · Decision Letter 2]

29 Oct 2021

Dear Dr Xiao,

Thank you for submitting your revised version of the Research Article entitled 'Bfc, a novel Serpent co-factor for the expression of Croquemort, regulates efferocytosis in Drosophila melanogaster' to PLOS Genetics.

The manuscript was fully evaluated again at the editorial level and by independent peer reviewers. The reviewers appreciated the attention to an important topic but identified some concerns that we ask you address in a revised manuscript

We therefore ask you to modify the manuscript according to the review recommendations. Your revisions should address the specific points made by each reviewer.

[LINK]

Yours sincerely,

Estee Kurant

Guest Editor

PLOS Genetics

Gregory P. Copenhaver

Editor-in-Chief

PLOS Genetics

Reviewer's Responses to Questions

**Comments to the Authors:**

Reviewer #1: The authors significantly improved their manuscript and addressed my main concerns.

I may have missed it but I didn't find an accession number to a public repository for the RNA-seq data. This should be provided.

Reviewer #2: The authors have satisfied all my comments and (in my opinion) have done a good job of answering the other reviewers's comments.

The manuscript is now far easier to read and the conclusions are more robust with the additional new data. The figures are much improved and more robust. Areas in which the paper was less strong have now been significantly improved (methods, presentation of gels/legends/descriptions, statistics).

Some typos remain and the manuscript should be proofed carefully. I list some errors I found while re-reading the manuscript to help the authors (see below); It may be that the editing company changed the meaning a little in some places.

This is really interesting work and I look forward to hearing about the other hits they have uncovered via this approach.

Figure 4B-B’’’ – I suggest authors add annotation to these panels to indicate S2 cell / S2 cell / wt mac / bfcKO mac to help the reader.

Line 308 – says authors generated the insertion line, when materials and methods says this was obtained from Bloomington.

Line 336 “Bfc act as…” should read “Bfc acts as…”

Line 490 Misspelling - Bfs rather than Bfc

Line 581 indicate rather than indicating

Line 783 TRITC rather than RITC?

Reviewer #3: The review is uploaded as an attachment.

**Have all data underlying the figures and results presented in the manuscript been provided?**

Reviewer #1: **No: **I may have missed it but I didn't find an accession number to a public repository for the RNA-seq data. This should be provided.

Reviewer #2: Yes

Reviewer #3: Yes

PLOS authors have the option to publish the peer review history of their article (what does this mean?). If published, this will include your full peer review and any attached files.

Reviewer #1: **Yes: **Lucas Waltzer

Reviewer #2: No

Reviewer #3: No

---

## [Decision Letter · Decision Letter 3]

15 Nov 2021

Dear Dr Xiao,

We are pleased to inform you that your manuscript entitled "bfc, a novel serpent co-factor for the expression of croquemort, regulates efferocytosis in Drosophila melanogaster" has been editorially accepted for publication in PLOS Genetics. Congratulations!

Yours sincerely,

Estee Kurant

Guest Editor

PLOS Genetics

Gregory P. Copenhaver

Editor-in-Chief

PLOS Genetics

Comments from the reviewers (if applicable):

Reviewer's Responses to Questions

**Comments to the Authors:**

Reviewer #1: The authors have complied with the reviewers comments. Remaining typos can certainly be dealt with at the proof stage.

Reviewer #3: The review is uploaded as an attachment.

**Have all data underlying the figures and results presented in the manuscript been provided?**

Reviewer #1: Yes

Reviewer #3: Yes

PLOS authors have the option to publish the peer review history of their article (what does this mean?). If published, this will include your full peer review and any attached files.

Reviewer #1: **Yes: **Lucas Waltzer

Reviewer #3: No

**Data Deposition**

http://datadryad.org/submit?journalID=pgenetics&manu=PGENETICS-D-21-00197R3

**Press Queries**

---

## [Editor Report · Acceptance letter]

29 Nov 2021

PGENETICS-D-21-00197R3 

*bfc*, a novel *serpent* co-factor for the expression of *croquemort*, regulates efferocytosis in *Drosophila melanogaster*

Dear Dr Xiao, 

We are pleased to inform you that your manuscript entitled "*bfc*, a novel *serpent* co-factor for the expression of *croquemort*, regulates efferocytosis in *Drosophila melanogaster*" has been formally accepted for publication in PLOS Genetics! Your manuscript is now with our production department and you will be notified of the publication date in due course.

With kind regards,

Agnes Pap

PLOS Genetics

On behalf of:
